# Graph Minimum Factorization Distance and Its Application to Large-Scale Graph Data Clustering

**Jicong Fan** [1]

## Abstract

Measuring the distance or similarity between graphs is the foundation of many graph analysis tasks, such as graph classification and clustering, but remains a challenge on large datasets. In this work, we treat the adjacency matrices of two graphs as two kernel matrices given by some unknown indefinite kernel function performed on two discrete distributions and define the distance between the two distributions as a measure, called MMFD, of the dissimilarity between two graphs. We show that MMFD is a pseudo-metric. Although the initial definition of MMFD seems complex, we show that it has a closed-form solution with extremely simple computation. To further improve the efficiency of large-scale clustering, we propose an MMFD-KM with linear space and time complexity with respect to the number of graphs. We also provide a generalization of MMFD, called MFD, which is more effective in exploiting the information of factors of adjacency matrices. The experiments on simulated graphs intuitively show that our methods are effective in comparing graphs. The experiments on real-world datasets demonstrate that, compared to the competitors, our methods have much better clustering performance in terms of three evaluation metrics and time cost.

## 1. Introduction

A graph is a collection of nodes (or vertices) connected by edges (or links). This structure is used to represent complex relationships and interactions between objects or entities. Common examples of graphs include chemical molecules (Rong et al., 2020), biology networks (Agarwal,

2006), social networks (Aggarwal, 2011), user-item interaction networks (Resnick & Varian, 1997), transportation networks (Bell & Iida, 1997), knowledge graphs (Hogan et al., 2021), and vision graphs (Riesen & Bunke, 2008). In graph data analysis, quantifying the similarity or dissimilarity between graphs (or comparing graphs for short) is probably the most fundamental and important problem and has numerous applications such as graph search, classification, and clustering. However, due to the non-Euclidean nature of graph data, comparing graphs remains a challenge especially when the graphs are large, though many techniques have been proposed in the past decades.

Numerous methods for comparing graphs are founded on the principles of graph isomorphism (Kobler et al., 2012) and its related concepts, including subgraph isomorphism and the identification of the largest common subgraph. Although the concept of examining graph isomorphism seems inherently straightforward, no efficient algorithms have been developed for this task. Moreover, graph isomorphism-based similarity measures and their derivatives require graphs to be precisely identical or to encompass substantial identical subgraphs to be considered similar, which is overly stringent (Shervashidze et al., 2011). As a result, many researchers attempted to develop more flexible measures of similarity. Representative examples of these more flexible measures include graph edit distance (Bunke, 1997; Neuhaus & Bunke, 2007; Abu-Aisheh et al., 2015; Chang et al., 2022), chemical distance (Jochum et al., 1980; Bento & Ioannidis, 2018), various graph kernels (Gärtner et al., 2003; Vishwanathan et al., 2010; Shervashidze et al., 2011; Nikolentzos et al., 2021), and Gromov-Wasserstein distance (Mémoli, 2011).

Graph edit distance (GED) is calculated by finding the minimum cost of a sequence of edit operations (e.g., node and edge insertion/deletion) that can transform one graph into another. While the computation of GED is NP-hard (Zeng et al., 2009), there are several approximation algorithms (Neuhaus & Bunke, 2007; Riesen & Bunke, 2009; Serratosa, 2014; 2015; Santacruz & Serratosa, 2020). For instance, Riesen & Bunke (2009) proposed a Bipartite algorithm that considers only local edge structure during the optimization process to address the high computational complexity of exact GED computation, meaning that the obtained distance

[1] School of Data Science, The Chinese University of Hong Kong, Shenzhen, China. Correspondence to: Jicong Fan <fanjicong@cuhk.edu.cn>.

*Proceedings of the 42nd International Conference on Machine Learning*, Vancouver, Canada. PMLR 267, 2025. Copyright 2025 by the author(s).

tends to be larger than the exact one. Most of these algorithms have a cubic computational cost with respect to the number of nodes (Serratosa, 2015). Moreover, one major limitation of GED is that it is not effective in comparing graphs with highly diverse sizes. For instance, given two complete graphs with very different numbers of nodes, the GED is very large, although the two graphs have similar topological patterns or practical functions.

Graph kernels are kernel functions that compute an inner product on graphs (Gärtner et al., 2003; Shervashidze et al., 2011; Mémoli, 2011; Bento & Ioannidis, 2018; Sun & Fan, 2024). Many graph kernels are based on the decomposition of graphs into a set of subgraphs or patterns, such as walks, paths, and trees, on which the similarity is measured. Examples of this kind of kernel include the shortest-path kernel (Borgwardt & Kriegel, 2005), graphlet kernel (Shervashidze et al., 2009), random walk kernel (Vishwanathan et al., 2010), etc. Other types of graph kernels include the neighborhood hash kernel (Hido & Kashima, 2009), Weisfeiler-Lehman (WL) kernel family (Shervashidze et al., 2011), multiscale Laplacian kernel (Kondor & Pan, 2016), Lovász $\vartheta$ kernel (Johansson et al., 2014), etc. Particularly, the WL kernel and its extensions (Kriege et al., 2016) are based on the Weisfeiler-Lehman test of isomorphism (Weisfeiler & Leman, 1968), making them very effective in comparing graphs. While most graph kernels on two graphs have at least cubic time complexity with respect to the number of nodes $n$ (Nikolentzos et al., 2021), the WL subtree kernel can be computed in time $\mathcal{O}(hl)$ (Shervashidze et al., 2011), where $h$ is the subtree height and $l$ is the number of edges of the graphs.

Recently, the Gromov-Wasserstein distance (GWD) (Mémoli, 2011) has shown promising performance in comparing graphs and the follow-up graph learning problems (Vayer et al., 2020; Xu et al., 2019; 2022). GWD is an extension of the Wasserstein distance (Villani et al., 2009; Cuturi & Doucet, 2014), which is used to compare probability distributions on a given metric space. GWD, on the other hand, allows for the comparison of probability distributions defined on different metric spaces. The time complexity of GWD is $\mathcal{O}(n^4)$ or $\mathcal{O}(n^3)$ (Peyré et al., 2016), which prohibits its applications to large graphs. Several attempts have been made to improve the scalability of GWD. For instance, Vayer et al. (2019) proposed sliced GW, with a complexity of $\mathcal{O}(n \log n)$. However, the sliced GW has limited applications because a feature representation is needed to allow the 1D projection. Kerdoncuff et al. (2021) proposed a sampled GW that uses the current estimate of the transport plan to guide the sampling of cost matrices, leading to a complexity of $\mathcal{O}(n^2)$. However, the sampled GW may suffer from accuracy degradation and inefficiency in real problems.

Besides the aforementioned direct comparison methods,

two graphs can be compared by first representing them as vectors using unsupervised learning and then calculating the similarity between the two vectors. There have been a few unsupervised graph representation learning (UGRL) methods that are based on graph neural networks (Kipf & Welling, 2017; Gilmer et al., 2017; Wu et al., 2024; Wang & Fan, 2024). Many of these methods (Sun et al., 2020; You et al., 2020; 2021; Shen et al., 2023) are based on the Info-Max principle (Linsker, 1988; Hjelm et al., 2019), where the mutual information between the graph-level representation and the representations of substructures of different scales (e.g., nodes and edges) are maximized to learn the neural network parameters. There are also UGRL methods based on other principles such as the Lovász principle proposed by (Sun et al., 2023). More methods of UGRL and more discussion can be found in (Sun et al., 2023). It is worth noting that there have been several end-to-end methods of clustering graphs (Ju et al., 2023; Cai et al., 2024), which also belong to UGRL, but can provide more discriminative representations. While graph kernels often rely on handcrafted features and have quadratic complexity with respect to the number of graphs $N$, UGRL methods can learn more flexible features of graphs and have a linear complexity with respect to $N$. Nevertheless, UGRL methods often have more hyperparameters (e.g., network size, learning rate, weights of terms in the loss function) to tune, which is never easy in practice. Moreover, they are still time-consuming in handling large graphs.

In this work, to effectively compare graphs and cluster graphs, we propose a novel distance measure, called minimum mean factorization distance (MMFD), and several extensions. The main idea behind MMFD is that the adjacency matrices of two graphs can be regarded as kernel matrices obtained by performing some unknown kernel function on two discrete distributions. Then, comparing the two distributions is a proxy for comparing the two graphs. Our contribution is summarized as follows.

- We present a novel distance measure MMFD between graphs, from the perspective of comparing distributions. We show that MMFD is a pseudo-metric and analyze its robustness to perturbations.
- We provide a low-rank approximation for MMFD and analyze the error bound theoretically. The low-rank MMFD is more efficient than MMFD.
- Based on MMFD, we develop a fast clustering method for datasets composed of a large number of graphs.
- We generalize MMFD to MFD, which is more effective in exploiting the information of adjacency matrices.
- Based on MFD, we present an efficient clustering algorithm for large datasets of graphs.

Table 1 shows the time complexities of two of our methods and some representative methods of comparing graphs. Our methods are very efficient, especially for a set of graphs.

Notably, besides the merit of efficiency, as shown in Section 3, our methods have much higher clustering accuracy in comparison to many strong competitors.

## 2. Methodology

### 2.1. Motivation

Let $G_1 = (V_1, E_1)$ and $G_2 = (V_2, E_2)$ be two undirected graphs in some space $\mathbb{G}$, where $V_i$ and $E_i$ with $i = 1$ or $2$ denote the node set and edge set respectively. We aim to provide a function

$$\text{dist} : \mathbb{G} \times \mathbb{G} \to \mathbb{R}$$

to quantify the distance between $G_1$ and $G_2$. We denote the self-looped adjacency matrices (possibly weighted) of $G_1$ and $G_2$ as $\mathbf{A}_1 \in \mathbb{R}^{n_1 \times n_1}$ and $\mathbf{A}_2 \in \mathbb{R}^{n_2 \times n_2}$, respectively.

Since the entries in an adjacency matrix quantify the affinity or similarity between data points or objects, it is reasonable to assume that $\mathbf{A}_1$ and $\mathbf{A}_2$ are generated by some kernel function $k$ on two sets of data points denoted as matrices $\mathbf{Z}_1 \in \mathbb{R}^{m \times n_1}$ and $\mathbf{Z}_2 \in \mathbb{R}^{m \times n_2}$ respectively, i.e.,

$$[\mathbf{A}_i]_{uv} = k(\mathbf{z}_u^{(i)}, \mathbf{z}_v^{(i)}), \quad i = 1, 2, \tag{1}$$

where $\mathbf{z}_u^{(i)}$ denotes the $u$-th column of $\mathbf{Z}_i$. For instance, in a social network $i$ with $n$ subjects, if subjects $u$ and $v$ are very similar to each other in terms of their attributes (denoted as $\mathbf{z}_u^{(i)}$ and $\mathbf{z}_v^{(i)}$), they are probably connected by an edge. The whole social network is based on a matrix $\mathbf{Z}_i$ of subjects' attributes. $\mathbf{Z}_1$ and $\mathbf{Z}_2$ can be regarded as two discrete distributions or two samples drawn from two unknown continuous distributions, respectively. Note that $k$ does not necessarily satisfy Mercer's condition (Ong et al., 2004; Hofmann et al., 2008) since $\mathbf{A}_1$ and $\mathbf{A}_2$ could be indefinite matrices. Here are some examples of $k$: $k_1(\mathbf{z}_u, \mathbf{z}_v) = (\mathbf{z}_u^\top \mathbf{z}_v + c)^2$, $k_2(\mathbf{z}_u, \mathbf{z}_v) = \exp(-\gamma \| \mathbf{z}_u - \mathbf{z}_v \|^2)$, $k_3(\mathbf{z}_u, \mathbf{z}_v) = \tanh(\mathbf{z}_u^\top \mathbf{z}_v - 1)$, $k_4(\mathbf{z}_u, \mathbf{z}_v) = \mathbb{1}(k_2(\mathbf{z}_u, \mathbf{z}_v) > \tau)$. Particularly, $k_3$ and $k_4$ (with appropriate $\tau$) are indefinite kernels, and $k_4$ can produce binary adjacency matrices.

To quantify the distance between $G_1$ and $G_2$, we propose to calculate the distance between $\mathbf{Z}_1$ and $\mathbf{Z}_2$ and let

$$\text{dist}(G_1, G_2) := f(\mathbf{Z}_1, \mathbf{Z}_2) \tag{2}$$

where $f : \mathbb{R}^{m \times n_1} \times \mathbb{R}^{m \times n_2} \to \mathbb{R}$ denotes a function to calculate the distance between two discrete distributions. Obviously, when $\mathbf{Z}_1$ and $\mathbf{Z}_2$ are identical (or up to a permutation), $G_1$ and $G_2$ are isomorphic, where $\text{dist}(G_1, G_2) = 0$. See the experiments in Appendix C.1.

### 2.2. Kernel Feature Map and PSD Matrix Construction

Suppose $G_1$ and $G_2$ have positive semi-definite (PSD) adjacency matrices $\mathbf{A}_1$ and $\mathbf{A}_2$ respectively, and denote the feature map induced by the PSD kernel $k$ as $\phi : \mathbb{R}^m \to \mathbb{R}^M$, where $M > m$. Then we have the following factorization (though not unique)

$$\mathbf{A}_i = \mathbf{\Phi}_i^\top \mathbf{\Phi}_i, \tag{3}$$

where $\mathbf{\Phi}_i = \phi(\mathbf{Z}_i) = [\phi(\mathbf{z}_1^{(i)}), \dots, \phi(\mathbf{z}_{n_i}^{(i)})]$, $i = 1, 2$. However, graphs with PSD adjacency matrices are rare in real applications, meaning that the factorization (3) often does not exist. To address the challenge, we need to generate some reasonable $\mathbf{\Phi}_i$ using $\mathbf{A}_i$, $i = 1, 2$, that can preserve the major information of $\mathbf{A}_1$ and $\mathbf{A}_2$, or in other words, find PSD proxies for $\mathbf{A}_1$ and $\mathbf{A}_2$.

In this work, we propose to use

$$\mathcal{A}_i^\phi = \sum_{j=1}^{n_i} |\lambda_j^{(i)}| \mathbf{v}_j^{(i)} \mathbf{v}_j^{(i)\top}, \quad i = 1, 2, \tag{4}$$

where $\lambda_j^{(i)}$ and $\mathbf{v}_j^{(i)}$ are the $j$-th eigenvalue and eigenvector of $\mathbf{A}_i$ (Luss & d'Aspremont, 2007), $i = 1, 2$. The motivation behind this is that we can decompose an indefinite kernel $k$ (Ong et al., 2004) as

$$k(\mathbf{z}, \mathbf{z}') = k^+(\mathbf{z}, \mathbf{z}') - k^-(\mathbf{z}, \mathbf{z}') \tag{5}$$

where both $k^+$ and $k^-$ are positive semi-definite. Denoting their feature maps as $\phi^+$ and $\phi^-$ respectively, we have $\mathbf{A}_i = \mathbf{\Phi}_i^{+\top} \mathbf{\Phi}_i^+ - \mathbf{\Phi}_i^{-\top} \mathbf{\Phi}_i^-$, $i = 1, 2$. Thus in (4), we have $\mathcal{A}_i^\phi = \mathbf{\Phi}_i^\top \mathbf{\Phi}_i$, where $\mathbf{\Phi}_i = [\mathbf{\Phi}_i^+; \mathbf{\Phi}_i^-]$, the row-direction concatenation of $\mathbf{\Phi}_i^+$ and $\mathbf{\Phi}_i^-$. Note that (4) is equivalent to $\mathcal{A}_i^\phi = \mathbf{U}_i \mathbf{S}_i \mathbf{U}_i^\top$, where $\mathbf{U}_i$ and $\mathbf{S}_i$ are the left (or right) singular vectors and singular values of $\mathbf{A}_i$ respectively.

### 2.3. Basic MMFD

Recall that in (2), we want to define an $f(\mathbf{Z}_1, \mathbf{Z}_2)$ to compare $\mathbf{Z}_1$ and $\mathbf{Z}_2$. However, both $\mathbf{Z}_1$ and $\mathbf{Z}_2$ are unknown. Instead, we know $\mathbf{\Phi}_1^\top \mathbf{\Phi}_1$ and $\mathbf{\Phi}_2^\top \mathbf{\Phi}_2$, where $\mathbf{\Phi}_1$ and $\mathbf{\Phi}_2$ are discrete distributions, given by

$$\mathcal{A}_i^\phi = \mathbf{\Phi}_i^\top \mathbf{\Phi}_i, \quad i = 1, 2. \tag{6}$$

Therefore, we propose to compare their mean vectors, i.e.,

$$\boldsymbol{\mu}_i := \frac{1}{n_i} \sum_{j=1}^{n_i} \phi(\mathbf{z}_j^{(i)}), \quad i = 1, 2. \tag{7}$$

Note that when $n_1 \neq n_2$, we can pad zero rows to the smaller one of $\mathbf{\Phi}_1$ and $\mathbf{\Phi}_2$ to ensure a consistent dimension $M$. Suppose $\phi$ can be well approximated by a $q$-order polynomial feature map, e.g., $\phi(\mathbf{z}) \approx [\dots, z_i, \dots, z_i z_j, \dots, z_i z_j \cdots z_k, \dots]^\top$, where the coefficients are omitted for convenience, then $\boldsymbol{\mu}_i$ can be regarded as an estimate of moments 1 to $q$ of the distribution. Thus,

| | $G_1, G_2$ | $G_1, G_2, \ldots, G_N$ |
|---|---|---|
| Shortest path kernel (Borgwardt & Kriegel, 2005) | $\mathcal{O}(n^4)$ | $\mathcal{O}(N^2 n^4)$ |
| Random walk kernel (Vishwanathan et al., 2010) | $\mathcal{O}(n^3)$ | $\mathcal{O}(N^2 n^3)$ |
| Weisfeiler-Lehman subtree kernel (Shervashidze et al., 2011) | $\mathcal{O}(hl)$ | $\mathcal{O}(Nhl + N^2 hn)$ |
| Graph Edit Distance (Serratosa, 2014) | $\mathcal{O}(n^3)$ | $\mathcal{O}(N^2 n^3)$ |
| (Entropic) Gromov–Wasserstein (Peyré et al., 2016) | $\mathcal{O}(n^3)$ | $\mathcal{O}(N^2 n^3)$ |
| Sampled Gromov–Wasserstein (Kerdoncuff et al., 2021) | $\mathcal{O}(n^2)$ | $\mathcal{O}(N^2 n^2)$ |
| MMFD$_{\text{LR}}$ | $\mathcal{O}(n^2 \log(d) + d^2 n)$ | $\mathcal{O}(N(n^2 \log(d) + d^2 n) + N^2)$ |
| MMFD$_{\text{LR}}$-KM | $\mathcal{O}(n^2 \log(d) + d^2 n)$ | $\mathcal{O}(N(n^2 \log(d) + d^2 n) + NKT)$ |

Table 1: Time complexity comparison between MMFD (with $d \ll n$) and a few representative graph distances or similarities on two graphs or a set of $N$ graphs, each with $n$ nodes. See Appendix C.4 for the running time comparison.

the distance between $\boldsymbol{\mu}_1$ and $\boldsymbol{\mu}_2$ is an effective measure of the dissimilarity between $\mathbf{Z}_1$ and $\mathbf{Z}_2$. However, the difficulty is that $\boldsymbol{\Phi}_1$ and $\boldsymbol{\Phi}_2$ are usually not in the same space since they cannot be uniquely determined by $\mathcal{A}_1^\phi$ and $\mathcal{A}_2^\phi$ (or $\mathbf{A}_1$ and $\mathbf{A}_2$) respectively. This means that $\boldsymbol{\mu}_1$ and $\boldsymbol{\mu}_2$ are not comparable. More precisely, for any orthonormal matrices $\mathbf{R}_1, \mathbf{R}_2 \in \mathbb{R}^{M \times M}$, letting

$$\bar{\boldsymbol{\Phi}}_i := \mathbf{R}_i \boldsymbol{\Phi}_i, \quad i = 1, 2, \tag{8}$$

we have $\bar{\boldsymbol{\Phi}}_i^\top \bar{\boldsymbol{\Phi}}_i = \boldsymbol{\Phi}_i^\top \mathbf{R}_i^\top \mathbf{R}_i \boldsymbol{\Phi}_i = \mathcal{A}_i^\phi$, meaning that $\bar{\boldsymbol{\Phi}}_i$ and $\boldsymbol{\Phi}_i$ are equivalent in terms of representing $\mathcal{A}_i^\phi$. Further discussion about this will be provided in Section 2.5.

To address the difficulty mentioned above, we propose aligning $\boldsymbol{\Phi}_1$ and $\boldsymbol{\Phi}_2$ using two projection matrices $\mathbf{R}_1$ and $\mathbf{R}_2$ to a common space so that we can compare the means of $\bar{\boldsymbol{\Phi}}_1$ and $\bar{\boldsymbol{\Phi}}_2$, denoted as $\bar{\boldsymbol{\mu}}_1$ and $\bar{\boldsymbol{\mu}}_2$ respectively, where $\bar{\boldsymbol{\mu}}_i = \mathbf{R}_i \boldsymbol{\mu}_i$, $i = 1, 2$. The question is what $\mathbf{R}_1$ and $\mathbf{R}_2$ we should choose. Given that we want to compare $\bar{\boldsymbol{\mu}}_1$ and $\bar{\boldsymbol{\mu}}_2$, a natural principle is that $\mathbf{R}_1$ and $\mathbf{R}_2$ make the distance between $\bar{\boldsymbol{\mu}}_1$ and $\bar{\boldsymbol{\mu}}_2$ as small as possible. Therefore, we may solve the following problem

$$\underset{\mathbf{R}_1, \mathbf{R}_2 \in \mathcal{R}}{\text{minimize}} \|\mathbf{R}_1 \boldsymbol{\mu}_1 - \mathbf{R}_2 \boldsymbol{\mu}_2\| \tag{9}$$

where $\mathcal{R}$ denotes the set of all real orthonormal matrices of size $M \times M$, i.e., $\mathcal{R} = \{\mathbf{R} \in \mathbb{R}^{M \times M} : \mathbf{R}^\top \mathbf{R} = \mathbf{I}_M\}$. In (9), we have $\|\mathbf{R}_1 \boldsymbol{\mu}_1 - \mathbf{R}_2 \boldsymbol{\mu}_2\| = \|\boldsymbol{\mu}_1 - \mathbf{R}_1^\top \mathbf{R}_2 \boldsymbol{\mu}_2\|$. It means that instead of optimizing two matrices, we can equivalently optimize a single matrix $\mathbf{R}_{12} := \mathbf{R}_1^\top \mathbf{R}_2$, subject to that $\mathbf{R}_{12} \in \mathcal{R}$. In other words, we can learn an $\mathbf{R}_{12}$ to align $\boldsymbol{\Phi}_2$ to the space of $\boldsymbol{\Phi}_1$, which simplifies the computation.

Based on (2), (9) and the corresponding analysis, letting

$$f(\mathbf{Z}_1, \mathbf{Z}_2) = \min_{\mathbf{R}_{12} \in \mathcal{R}} \|\boldsymbol{\mu}_1 - \mathbf{R}_{12} \boldsymbol{\mu}_2\| \tag{10}$$

we achieve the following distance between two graphs.

**Definition 2.1** (MMFD). Let $\mathcal{A}_i^\phi = \boldsymbol{\Phi}_i^\top \boldsymbol{\Phi}_i$ and denote the $j$-th column of $\boldsymbol{\Phi}_i$ as $\phi(\mathbf{z}_j^{(i)})$, $i = 1, 2$. The minimum mean

factorization distance between $G_1$ and $G_2$ is defined as

$$\begin{aligned} &\text{MMFD}(G_1, G_2) \\ &= \min_{\mathbf{R}_{12} \in \mathcal{R}} \left\| \frac{1}{n_1} \sum_{j=1}^{n_1} \phi(\mathbf{z}_j^{(1)}) - \frac{1}{n_2} \sum_{j=1}^{n_2} \mathbf{R}_{12} \phi(\mathbf{z}_j^{(2)}) \right\| \end{aligned} \tag{11}$$

We provide the following theory for MMFD.

**Theorem 2.2** (Pseudo-metric). *Let $\mathbb{G}$ be the set of all undirected graphs. MMFD satisfies the following axioms.*
*(a) $MMFD(G_1, G_2) \geq 0$, $MMFD(G_1, G_2) = MMFD(G_2, G_1)$, and $MMFD(G_1, G_1) = 0$ hold for any $G_1, G_2$ in $\mathbb{G}$;*
*(b) $MMFD(G_1, G_2) \leq MMFD(G_1, G_3) + MMFD(G_2, G_3)$ holds for any $G_1, G_2, G_3$ in $\mathbb{G}$.*

The theorem shows that MMFD is a pseudo-metric, where $\text{MMFD}(G_1, G_2) = 0 \iff G_1 = G_2$ cannot be theoretically guaranteed. For instance:

**Proposition 2.3.** *For any two complete graphs $G_1$ and $G_2$, whenever $n_1 = n_2$ or $n_1 \neq n_2$, $MMFD(G_1, G_2) \equiv 0$.*

The following theorem shows that MMFD has a closed-form solution, which is extremely simple. Although in the motivation, definition, and optimization, we used $\boldsymbol{\Phi}_i, \mathbf{Z}_i$, $i = 1, 2$, $\mathbf{R}_{12}$, etc., they are not present in the final format of MMFD.

**Theorem 2.4.** *Based on Definition 2.1, it follows that*

$$MMFD(G_1, G_2) = \left| \frac{1}{n_1} \sqrt{\sum_{uv} [\mathcal{A}_1^\phi]_{uv}} - \frac{1}{n_2} \sqrt{\sum_{uv} [\mathcal{A}_2^\phi]_{uv}} \right| \tag{12}$$

As a distance measure between graphs, it is important to understand the robustness of MMFD to perturbation on the adjacency matrices. Hence, we present the following result.

**Theorem 2.5** (Robustness). *Given $G_1, G_2$, assume $n_1 = n_2 = n$ and let $\boldsymbol{\Delta}_1, \boldsymbol{\Delta}_2 \in \mathbb{R}^{n \times n}$ be perturbations on $\mathbf{A}_1, \mathbf{A}_2$ respectively, resulting in perturbed graphs $G_1', G_2'$*

*respectively, where $\sigma_1(\mathbf{\Delta}_i) \le \frac{1}{2}\sigma_1(\mathbf{A}_i)$, $i = 1, 2$. Then*

$$\left| MMFD(G_1', G_2') - MMFD(G_1, G_2) \right|$$
$$\le \frac{5}{n^{1/2}}\left( \delta_1^{-1}\sigma_1(\mathbf{A}_1)\sigma_1(\mathbf{\Delta}_1) + \delta_2^{-1}\sigma_1(\mathbf{A}_2)\sigma_1(\mathbf{\Delta}_2) \right)^{1/2}$$

*where $\delta_i = \min_{j \ne k} |\sigma_j(\mathbf{A}_i) - \sigma_k(\mathbf{A}_i)|$, $i = 1, 2$.*

The theorem shows that when the graphs are perturbed slightly and the original adjacency matrices have distinct singular values, MMFD does not change significantly. This is important for downstream tasks where MMFDs between graphs in the same cluster should be small.

Figure 2 shows the MMFD matrix on seven small graphs. We can see the distance measures are consistent with our intuitive observation on the graphs. For instance, $G_2$ is more similar to $G_3$ than to $G_1$; $G_7$ lies between $G_5$ and $G_6$; the difference between $G_6$ and $G_7$ is less than the difference between $G_5$ and $G_6$. In sum, MMFD is effective in comparing graphs. Moreover, according to Table 10 of Appendix C.6, MMFD can outperform other methods.

| | | | | | | |
|---|---|---|---|---|---|---|
| – | **0.0914** | 0.1589 | 0.2097 | 0.2528 | 0.2505 | 0.2505 |
| 0.0914 | – | **0.0675** | 0.1182 | 0.1614 | 0.1590 | 0.1591 |
| 0.1589 | 0.0675 | – | **0.0507** | 0.0939 | 0.0915 | 0.0916 |
| 0.2097 | 0.1182 | 0.0507 | – | 0.0432 | **0.0408** | 0.0409 |
| 0.2528 | 0.1614 | 0.0939 | 0.0432 | – | 0.0024 | **0.0023** |
| 0.2505 | 0.1590 | 0.0915 | 0.0408 | 0.0024 | – | **0.0001** |
| 0.2505 | 0.1591 | 0.0916 | 0.0409 | 0.0023 | **0.0001** | – |

Table 2: MMFDs between seven synthetic graphs $(G_1, G_2, \ldots, G_7$ from left to right). In each row of the matrix, the values of distances to the closest graph and most distant graph are highlighted in red and blue respectively.

### 2.4. Low-Rank MMFD

Currently, the time complexity of MMFD is $\mathcal{O}(n^3)$ with $n_1 = n_2 = n$ assumed, mainly due to the eigenvalue decomposition (EVD) or singular value decomposition (SVD) when constructing the PSD adjacency matrices, though it can be lowered when taking advantage of the sparsity of the adjacency matrices. For large graphs, instead of the full SVD, we perform randomized SVD (Halko et al., 2011) on $\mathbf{A}_1$ and $\mathbf{A}_2$ to obtain the following low-rank approximation

$$\mathbf{A}_i \approx \bar{\mathbf{U}}_i\bar{\mathbf{S}}_i\bar{\mathbf{V}}_i^\top, \quad i = 1, 2, \tag{13}$$

where $\bar{\mathbf{U}}_i, \bar{\mathbf{V}}_i \in \mathbb{R}^{n \times d}$, $\bar{\mathbf{S}}_i \in \mathbb{R}^{d \times d}$, and $d$ is much less than $n$. Now for (11), we let $\mathbf{\Phi}_i = \bar{\mathbf{S}}_i^{1/2}\bar{\mathbf{V}}_i^\top$. Letting

$\bar{\mathcal{A}}_i^\phi = \bar{\mathbf{V}}_i\bar{\mathbf{S}}_i\bar{\mathbf{V}}_i^\top$, we obtain the following low-rank MMFD

$$\text{MMFD}_{\text{LR}}(G_1, G_2) = \frac{1}{n}\left| \sqrt{\sum_{uv}[\bar{\mathcal{A}}_1^\phi]_{uv}} - \sqrt{\sum_{uv}[\bar{\mathcal{A}}_2^\phi]_{uv}} \right| \tag{14}$$

which can be extended to the case that $n_1 \ne n_2$. The time complexity of $\text{MMFD}_{\text{LR}}$ is $\mathcal{O}(n^2\log(d) + d^2 n)$ mainly caused by the randomized SVD (Halko et al., 2011) of $\mathbf{A}_i$. Note that $\text{MMFD}_{\text{LR}}$ is also a pseudo-metric. The proof is similar to that for MMFD and hence is omitted.

The following theorem shows the bound of the difference between MMFD and $\text{MMFD}_{\text{LR}}$.

**Theorem 2.6.** *Suppose $n_1 = n_2 = n$ and denote $\sigma_1(\mathbf{A}_i) \ge \sigma_2(\mathbf{A}_i) \ge \cdots \ge \sigma_n(\mathbf{A}_i)$ the singular values of $\mathbf{A}_i$, $i = 1, 2$. It holds that*

$$\left| MMFD_{LR}(G_1, G_2) - MMFD(G_1, G_2) \right|$$
$$\le \frac{1}{n^{1/2}}\left( \sum_{j=d+1}^{n}\left( \sigma_j(\mathbf{A}_1) + \sigma_j(\mathbf{A}_2) \right) \right.$$
$$\left. + 2\left( \sigma_1^{1/2}(\mathbf{A}_1)\sigma_{d+1}^{1/2}(\mathbf{A}_2) + \sigma_1^{1/2}(\mathbf{A}_2)\sigma_{d+1}^{1/2}(\mathbf{A}_1) \right) \right)^{1/2}$$

We see that when $\sigma_{d+1}(\mathbf{A}_i)$ are smaller, the bound becomes tighter. Indeed, the adjacency matrices of large graphs are often approximately low-rank, meaning that $\sigma_{d+1}(\mathbf{A}_i)$ are small. Although leading to some information loss, $\text{MMFD}_{\text{LR}}$ not only improves the computational efficiency but also has the effect of denoising because smaller singular values often correspond to the noise in data.

Figure 1 compares $\text{MMFD}_{\text{LR}}$ of different rank with MMFD on the seven graphs introduced in Table 2. $\text{MMFD}_{\text{LR}}$ is more effective in distinguishing highly similar graphs.

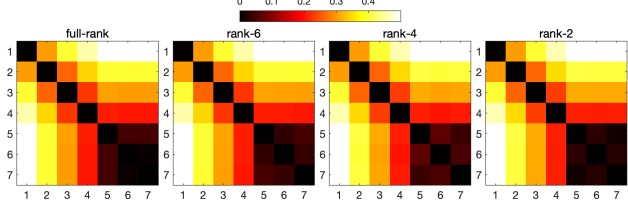

Figure 1: Visualization of $\text{MMFD}_{\text{LR}}$ (with different rank $d$) between the seven graphs in Table 2. For clearer visualization, we showed $\text{MMFD}_{\text{LR}}^{1/2}$.

### 2.5. Comparison with MMD

MMFD is closely related to the maximum mean discrepancy (MMD) (Gretton et al., 2012), though the latter is not related to graphs. Performing MMD on $\mathbf{Z}_1$ and $\mathbf{Z}_2$ using $\mathbf{\Phi}_1$ and $\mathbf{\Phi}_2$ directly, we have $\text{MMD}(\mathbf{Z}_1, \mathbf{Z}_2) =$

$\frac{1}{n}\sqrt{\sum_{uv}[\boldsymbol{\mathcal{A}}_1^\phi]_{uv} + \sum_{uv}[\boldsymbol{\mathcal{A}}_2^\phi]_{uv} - 2\langle\mathbf{H}, \boldsymbol{\Phi}_1^\top\boldsymbol{\Phi}_2\rangle}$, where $\mathbf{H}$ is an $n \times n$ matrix consisting of ones only and we have assumed $n_1 = n_2 = n$ for simplicity. We see that compared with MMFD, MMD($\mathbf{Z}_1, \mathbf{Z}_2$) does not involve optimizing an $\mathbf{R}_{12}$. Essentially, in (11), when we fix $\mathbf{R}_{12}$ as an identity matrix, MMFD degrades to MMD. The fundamental limitation of MMD($\mathbf{Z}_1, \mathbf{Z}_2$) is that it makes no sense when $\boldsymbol{\Phi}_1$ and $\boldsymbol{\Phi}_2$ are not in the same space (i.e., the features in $\boldsymbol{\Phi}_1$ and $\boldsymbol{\Phi}_2$ are not comparable), which is always the case and is explained as follows.

- In EVD (or SVD), though well-established, individual eigenvectors (or singular vectors) have arbitrary signs (Bro et al., 2008). This sign ambiguity in $\boldsymbol{\Phi}_1$ and $\boldsymbol{\Phi}_2$ makes MMD failed to quantify the dissimilarity between $\boldsymbol{\Phi}_1$ and $\boldsymbol{\Phi}_2$.

- When $\mathbf{A}_i$ has repeated eigenvalues, eigenvectors sharing a common eigenvalue cannot be determined uniquely even if the sign ambiguity is ignored. The reason is that any set of orthogonal vectors lying in their span are also eigenvectors with that eigenvalue. This means $\boldsymbol{\Phi}_1$ and $\boldsymbol{\Phi}_2$ cannot be compared by MMD.

- Sorting eigenvectors according to eigenvalues may not be optimal in comparing the adjacency matrices, especially when the adjacent eigenvalues are similar.

### 2.6. MMFD with Node Attributes

In many applications, the nodes of graphs have inherent attributes, which provide additional information and should be exploited effectively. Let $\mathbf{X}_i \in \mathbb{R}^{l \times n_i}$ be the matrix of nodes' attributes of $G_i$, $i = 1, 2$. We may consider augmenting MMFD or MMFD$_{\text{LR}}$ with MMD($\mathbf{X}_1, \mathbf{X}_2$). However, this neglects the correspondence between each node and its attribute vector, which can cause the inability to distinguish between $G_1$ and $G_2$. To address this problem, we propose to use some technique (e.g. k-nearest neighbor) to construct two graphs (denoted as $G_1', G_2'$) from $\mathbf{X}_1$ and $\mathbf{X}_2$ respectively and append $G_1', G_2'$ to the original graphs to obtain two larger graphs $\tilde{G}_1, \tilde{G}_2$, where the node $j$ of $G_i'$ is connected to the node $j$ of $G_i$. The main idea is

$$\tilde{G}_i = \text{APPEND}\big(G_i, \text{G-CONSTRUCT}(\mathbf{X}_i)\big) \quad (15)$$

Then we calculate MMFD($\tilde{G}_1, \tilde{G}_2$) or MMFD$_{\text{LR}}(\tilde{G}_1, \tilde{G}_2)$. For instance, the top of Figure 2 shows two simple graphs with the same adjacency matrix but different node attributes. In this case, MMFD($G_1, G_2$) = 0 and MMD($\mathbf{X}_1, \mathbf{X}_2$) = 0, but $G_1$ and $G_2$ are different. In contrast, as shown by the bottom of Figure 2, $\tilde{G}_1$ and $\tilde{G}_2$ are different and we have MMFD($\tilde{G}_1, \tilde{G}_2$) = 0.0026 and MMFD$_{\text{LR}}(\tilde{G}_1, \tilde{G}_2)$ = 0.0032 ($d = 3$). It is worth mentioning that, for MMFD, compared to (15), there may exist more effective strategies

for exploiting node attributes, e.g., using graph neural networks.

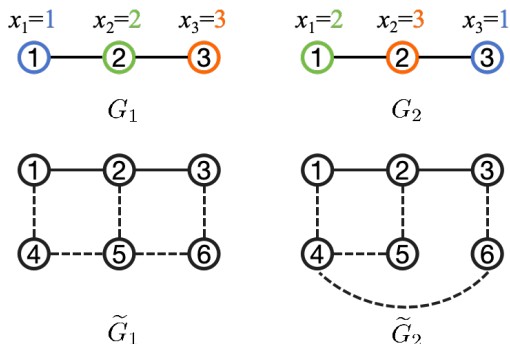

Figure 2: Integration of adjacency matrix and node attributes. $G_1$ and $G_2$ are two graphs with the same adjacency matrix but different node attributes (colored), meaning that they are not equal. $\tilde{G}_1$ and $\tilde{G}_2$ are the integrated graphs (without node attributes) using the original adjacency matrices and the 1-NN graphs constructed from the nodes' attributes. Nodes 4, 5, 6 correspond to the attributes $x_1$, $x_2$, $x_3$, respectively. $\tilde{G}_1$ and $\tilde{G}_2$ are different (not isomorphic).

For convenience, we show the computation steps of MMFD and MMFD$_{\text{LR}}$ in Algorithm 1. As a distance measure between graphs, MMFD is useful in many problems, such as graph classification, clustering, and visualization. We can define a graph kernel based on MMFD as follows

$$k(G_1, G_2) = \exp\big(-\gamma \times \text{MMFD}^2(G_1, G_2)\big) \quad (16)$$

where $\gamma > 0$ is a hyperparameter. With this kernel, we can conduct kernel density estimation, support vector machine (Cortes & Vapnik, 1995), spectral clustering (Ng et al., 2001), etc.

---

**Algorithm 1** Computation of MMFD between Graphs

**Input:** Graphs $G_1, G_2$ with self-looped adjacency matrices $\mathbf{A}_1, \mathbf{A}_2$ and node attribute matrices $\mathbf{X}_1, \mathbf{X}_2$ (if available) respectively, *using low-rank ($d$)*.
1: **if** $\mathbf{X}_1, \mathbf{X}_2$ are available and should be used **then**
2:     Construct graph from $\mathbf{X}_i$ and append to $G_i$ to generate an augmented $\mathbf{A}_i$, $i = 1, 2$.
3: **end if**
4: SVD or randomized SVD: $\mathbf{A}_i \approx \mathbf{U}_i\mathbf{S}_i\mathbf{V}_i^\top$, $i = 1, 2$.
5: **if** *using low-rank* is True **then**
6:     PSD: $\boldsymbol{\mathcal{A}}_i^\phi = \bar{\mathbf{V}}_i\bar{\mathbf{S}}_i\bar{\mathbf{V}}_i^\top$, $i = 1, 2$. ($\bar{\mathbf{V}}_i = [\mathbf{v}_{i1}, \dots, \mathbf{v}_{id}]$)
7: **else**
8:     PSD: $\boldsymbol{\mathcal{A}}_i^\phi = \mathbf{V}_i\mathbf{S}_i\mathbf{V}_i^\top$, $i = 1, 2$.
9: **end if**
10: MMFD $= \left|\frac{1}{n_1}\sqrt{\sum_{uv}[\boldsymbol{\mathcal{A}}_1^\phi]_{uv}} - \frac{1}{n_2}\sqrt{\sum_{uv}[\boldsymbol{\mathcal{A}}_2^\phi]_{uv}}\right|$
**Output:** MMFD

---

## 2.7. MMFD for Large-Scale Clustering

Given a set of $N$ graphs, denoted as $\mathcal{G} = \{G_1, G_2, \ldots, G_N\}$, belonging to $K$ classes, if we perform spectral clustering with MMFD$_{\text{LR}}$ on $\mathbb{G}$, the space complexity and time complexity are quadratic with $N$, which prevent its applications to very large graph datasets.

Inspired by K-means clustering, we may consider finding $K$ graphs, $\bar{G}_1, \ldots, \bar{G}_K$, representing the centers of clusters $\mathcal{G}_1, \ldots, \mathcal{G}_K$ respectively, such that the sum of the squared distances measure by MMFD$_{\text{LR}}$ (or MMFD) between the graphs and their corresponding centers is minimum, i.e.,

$$\underset{\bar{G}_1, \ldots, \bar{G}_K}{\text{minimize}} \sum_{j=1}^{K} \sum_{G_i \in \mathcal{G}_j} \text{MMFD}^2_{\text{LR}}(G_i, \bar{G}_j) \qquad (17)$$

However, we cannot obtain $\bar{G}_1, \ldots, \bar{G}_K$ themselves because MMFD$_{\text{LR}}$ is operated on the PSD adjacency matrices. Actually, we do not need to learn $\bar{G}_1, \ldots, \bar{G}_K$. Based on (14), we can rewrite (17) as

$$\underset{\bar{G}_1, \ldots, \bar{G}_K}{\text{minimize}} \sum_{j=1}^{K} \sum_{G_i \in \mathcal{G}_j} \left| \frac{1}{n_i} \sqrt{\sum_{uv} [\bar{\mathcal{A}}_i^{\phi}]_{uv}} - \frac{1}{n} \sqrt{\sum_{uv} [\mathbf{B}_j]_{uv}} \right|^2 \qquad (18)$$

where $\mathbf{B}_j$ and $n$ denote the adjacency matrix and the number of nodes of $\bar{G}_j$, $j = 1, \ldots, K$. Letting $g_i = \frac{1}{n_i} \sqrt{\sum_{uv} [\bar{\mathcal{A}}_i^{\phi}]_{uv}}$ and $c_j = \frac{1}{n} \sqrt{\sum_{uv} [\mathbf{B}_j]_{uv}}$, we obtain from (18) that

$$\underset{c_1, \ldots, c_K}{\text{minimize}} \sum_{j=1}^{K} \sum_{G_i \in \mathcal{G}_j} (g_i - c_j)^2 \qquad (19)$$

where $\mathcal{G}_j = \{G : (g - c_j)^2 < (g - c_l)^2 \; \forall l \neq j\}$. This is actually K-means on $N$ 1-D data points, where we only need to optimize scalars $c_1, \ldots, c_K$, rather than $\bar{G}_1, \ldots, \bar{G}_K$. Suppose $n_1 = \ldots = n_N = n$, then the total space and time complexity are $\mathcal{O}(n^2 + dn + N + K)$ and $\mathcal{O}(N(n^2 \log(d) + d^2 n) + NKT)$, both linear with $N$, where $T$ denotes the number of iterations in K-means. For convenience, we call this clustering method MMFD-KM.

## 2.8. MFD–Beyond Mean Comparison

MMFD compares only the mean vectors of $\mathbf{\Phi}_1$ and $\mathbf{\Phi}_2$. To exploit more information of $\mathbf{\Phi}_1$ and $\mathbf{\Phi}_2$, we provide a generalized method, called MFD, in the following.

**Definition 2.7** (MFD). Let $\mathcal{A}_i^{\phi} = \mathbf{\Phi}_i^{\top} \mathbf{\Phi}_i$, $i = 1, 2$. The minimum factorization distance between $G_1$ and $G_2$ is defined as

$$\text{MFD}(G_1, G_2)$$
$$= \min_{\mathbf{R}_{12} \in \mathcal{R}} \left\| \frac{1}{n_1} \sum_{j=1}^{n_1} \zeta(\phi_j^{(1)}) - \frac{1}{n_2} \sum_{j=1}^{n_2} \zeta(\mathbf{R}_{12} \phi_j^{(2)}) \right\|$$

$$= \min_{\mathbf{R}_{12} \in \mathcal{R}} \left( \frac{1}{n_1^2} \sum_{u,v} k(\phi_u^{(1)}, \phi_v^{(1)}) + \frac{1}{n_2^2} \sum_{u,v} k(\mathbf{R}_{12} \phi_u^{(2)}, \mathbf{R}_{12} \phi_v^{(2)}) \right.$$
$$\left. - \frac{2}{n_1 n_2} \sum_{u,v} k(\phi_u^{(1)}, \mathbf{R}_{12} \phi_v^{(2)}) \right)^{1/2} \qquad (20)$$

where $k$ denotes a nonlinear kernel with feature map $\zeta$.

The following theorem shows that MFD is a pseudo-metric under some mild assumptions.

**Theorem 2.8.** *When the kernel function in* (20) *is a rotation-invariant kernel (e.g., radial basis function kernel), MFD is a pseudo-metric.*

Unlike MMFD, MFD has no closed-form solution and has a higher computational cost. Letting $k$ in (20) be the Gaussian kernel with parameter $\beta$, we have

$$\text{MFD}(G_1, G_2)$$
$$= \min_{\mathbf{R}_{12} \in \mathcal{R}} \left( \frac{1}{n_1^2} \sum_{u,v} \exp(-\beta \|\phi_u^{(1)} - \phi_v^{(1)}\|^2) \right.$$
$$+ \frac{1}{n_2^2} \sum_{u,v} \exp(-\beta \|\phi_u^{(2)} - \phi_v^{(2)}\|^2)$$
$$\left. - \frac{2}{n_1 n_2} \sum_{u,v} \exp(-\beta \|\phi_u^{(1)} - \mathbf{R}_{12} \phi_v^{(2)}\|^2) \right)^{1/2}$$
$$= \min_{\mathbf{R}_{12} \in \mathcal{R}} \left( C - \frac{2}{n_1 n_2} \sum_{u,v} \exp\left( -\beta \left( [\mathcal{A}_1^{\phi}]_{uu} + [\mathcal{A}_2^{\phi}]_{vv} \right.\right.\right.$$
$$\left.\left.\left. - 2\langle \phi_u^{(1)}, \mathbf{R}_{12} \phi_v^{(2)} \rangle \right) \right) \right)^{1/2} \qquad (21)$$

where $C = \frac{1}{n_1^2} \sum_{u,v} \exp\left( -\beta([\mathcal{A}_1^{\phi}]_{uu} + [\mathcal{A}_1^{\phi}]_{vv} - 2[\mathcal{A}_1^{\phi}]_{uv}) \right) + \frac{1}{n_2^2} \sum_{u,v} \exp\left( -\beta([\mathcal{A}_2^{\phi}]_{uu} + [\mathcal{A}_2^{\phi}]_{vv} - 2[\mathcal{A}_2^{\phi}]_{uv}) \right)$. Thus, for MFD, we need to solve the following problem

$$\underset{\mathbf{R}_{12} \in \mathcal{R}}{\text{maximize}} \sum_{u,v} \alpha_{uv} \exp\left( 2\beta \langle \phi_u^{(1)}, \mathbf{R}_{12} \phi_v^{(2)} \rangle \right) \triangleq \mathcal{L}(\mathbf{R}_{12}) \qquad (22)$$

where $\alpha_{uv} = \exp(-\beta([\mathcal{A}_1^{\phi}]_{uu} + [\mathcal{A}_2^{\phi}]_{vv}))$. This is a challenging problem and has no closed-form solution because of the sum-of-exp form and the orthonormality constraint.

To tackle the difficulty, letting $\mathbf{R}_{12}^{(t-1)}$ be the solution at iteration $t - 1$, we consider the following quadratic approximation at iteration $t$

$$\mathcal{L}(\mathbf{R}_{12}) \approx \mathcal{L}(\mathbf{R}_{12}^{(t-1)}) + \langle \mathbf{G}^{(t-1)}, \mathbf{R}_{12} - \mathbf{R}_{12}^{(t-1)} \rangle$$
$$- \frac{\rho}{2} \|\mathbf{R}_{12} - \mathbf{R}_{12}^{(t-1)}\|_F^2 \qquad (23)$$

where $\mathbf{G}^{(t-1)} = \nabla \mathcal{L}(\mathbf{R}_{12}^{(t-1)}) = 2\beta \sum_{uv} \alpha_{uv} \exp(2\beta \langle \phi_u^{(1)}, \mathbf{R}_{12}^{(t-1)} \phi_v^{(2)} \rangle) \phi_u^{(1)} \phi_v^{(2)\top}$ and

$\rho > 0$ is a hyperparameter to be determined. Now we solve the following problem at iteration $t$:

$$\underset{\mathbf{R}_{12} \in \mathcal{R}}{\text{maximize}} \ \mathcal{L}(\mathbf{R}_{12}^{(t-1)}) + \langle \mathbf{G}^{(t-1)}, \mathbf{R}_{12} - \mathbf{R}_{12}^{(t-1)} \rangle \\ - \frac{\rho}{2} \|\mathbf{R}_{12} - \mathbf{R}_{12}^{(t-1)}\|_F^2 \qquad (24)$$

or the following equivalently

$$\underset{\mathbf{R}_{12} \in \mathcal{R}}{\text{maximize}} \ \langle \mathbf{G}^{(t-1)} + \rho \mathbf{R}_{12}^{(t-1)}, \mathbf{R}_{12} \rangle \qquad (25)$$

Letting $\mathbf{Q}_t = \mathbf{G}^{(t-1)} + \rho \mathbf{R}_{12}^{(t-1)}$, we rewrite (25) as

$$\underset{\mathbf{R}_{12} \in \mathcal{R}}{\text{maximize}} \ \langle \mathbf{R}_{12}, \mathbf{Q}_t \rangle \qquad (26)$$

Then we have the following closed-form solution

$$\mathbf{R}_{12}^{(t)} = \mathbf{U}\mathbf{V}^\top, \qquad (27)$$

where $\mathbf{U}$ and $\mathbf{V}$ are from the SVD of $\mathbf{Q}_t$, i.e., $\mathbf{Q}_t = \mathbf{U}\mathbf{S}\mathbf{V}^\top$.

We summarize the optimization steps for MFD in Algorithm 2, where the utilization of node attributes (if available) is omitted for simplicity. The following theorem provides a theoretical guarantee of the convergence of the optimization.

**Theorem 2.9.** *Let $\{\mathcal{L}(\mathbf{R}_{12}^{(t)})\}_t$ and $\{\mathbf{R}_{12}^{(t)}\}_t$ be the sequences given by Algorithm 2. For any $\rho > 0$, it holds that:*
*(a) $\{\mathcal{L}(\mathbf{R}_{12}^{(t)})\}_t$ is non-decreasing, i.e., $\mathcal{L}(\mathbf{R}_{12}^{(t)}) \geq \mathcal{L}(\mathbf{R}_{12}^{(t-1)}) + \frac{\rho}{2}\|\mathbf{R}_{12}^{(t)} - \mathbf{R}_{12}^{(t-1)}\|_F^2$;*
*(b) $\{\mathbf{R}_{12}^{(t)}\}_t$ is convergent, i.e., $\mathbf{R}_{12}^{(t)} - \mathbf{R}_{12}^{(t-1)} \rightarrow \mathbf{0}$ when $t \rightarrow \infty$.*

The time complexity of MFD with low-rank approximation is $\mathcal{O}(n^2 \log(d) + d^2 n + T(d^3 + dn^2))$, which is much higher than that of MMFD$_{\text{LR}}$. However, in the experiments (see Table 3 and Figure 11), the performance of MFD is better than that of MMFD, because MFD can utilize more information of $\mathbf{\Phi}_1$ and $\mathbf{\Phi}_2$.

## 2.9. MFD for Large-Scale Clustering

Based on MFD, we provide a fast clustering algorithm called MFD-KD, which is detailed in Appendix A.2.

## 3. Experiments

Previous studies usually used graph classification to evaluate graph kernels or distances. Differently, in this work, we use clustering, due to the following reasons. First, clustering is more challenging than classification, especially for imbalanced datasets. Second, clustering does not involve data splitting and has fewer hyperparameters to tune, leading

---

**Algorithm 2** Computation of MFD between Graphs

**Input:** Graphs $G_1, G_2$ with self-looped adjacency matrices $\mathbf{A}_1, \mathbf{A}_2, d, \beta, \varepsilon, T, \rho$.
1: Randomized SVD ($d$): $\mathbf{A}_i \approx \bar{\mathbf{U}}_i \bar{\mathbf{S}}_i \bar{\mathbf{V}}_i^\top$, $i = 1, 2$.
2: $\mathbf{\Phi}_i = \bar{\mathbf{S}}_i^{1/2} \bar{\mathbf{V}}_i^\top$, $i = 1, 2$.
3: Initialization: $\mathbf{R}_{12}^{(0)} = \mathbf{I}_d$, $t = 0$.
4: **while** $t < T$ **do**
5:     $t \leftarrow t + 1$.
6:     $\mathbf{Q}_t = \Big( 2\beta \sum_{uv} \alpha_{uv} \exp \big( 2\beta \langle \phi_u^{(1)}, \mathbf{R}_{12}^{(t-1)} \phi_v^{(2)} \rangle \big) \times$
         $\phi_u^{(1)} \phi_v^{(2)\top} \Big) + \rho \mathbf{R}_{12}^{(t-1)}$
7:     Perform SVD: $\mathbf{Q}_t = \mathbf{U}\mathbf{S}\mathbf{V}^\top$.
8:     $\mathbf{R}_{12}^{(t)} = \mathbf{U}\mathbf{V}^\top$.
9:     **if** $\|\mathbf{R}_{12}^{(t)} - \mathbf{R}_{12}^{(t-1)}\|_F / \sqrt{d} < \varepsilon$ **then**
10:       Break.
11:     **end if**
12: **end while**
13: Calculate MFD using $\mathbf{R}_{12}^{(t)}$ and (21).
**Output:** MFD$(G_1, G_2)$

---

to lower uncertainty. Finally, our MMFD-KM and MFD-KD are specified for clustering. We consider six large or relatively large datasets of graphs from the TUDatasets (Morris et al., 2020). More details are in Appendix B.1.

The proposed methods are compared with three categories of competitors. The first category is graph kernels including SP (Borgwardt & Kriegel, 2005), GK (Shervashidze et al., 2009), RW (Vishwanathan et al., 2010), WL (Shervashidze et al., 2011), LT (Johansson et al., 2014), and WL-OA (Kriege et al., 2016). The second category is unsupervised graph representation learning methods including InfoGraph (Sun et al., 2020), GraphCL (You et al., 2020), JOAO (You et al., 2021), and GWF (Xu et al., 2022), followed by K-means or spectral clustering. The last category is the end-to-end GNN-based graph-level clustering methods including GLCC (Ju et al., 2023) and DCGLC (Cai et al., 2024). Besides, we also include MMD (detailed in Section 2.5), Gromov- Wasserstein distance (GWD) with entropic regularization, and graph edit distance (GED) (Serratosa, 2014) in the experiments. More details about the settings are in Appendix B.2. The evaluation metrics for clustering performance are clustering accuracy (ACC), normalized mutual information (NMI), and adjusted rand index (ARI).

Table 3 shows the clustering results on the four datasets with moderate sizes, where for all compared methods except MMD, GWD, and GED, we just use the results reported in the paper of (Cai et al., 2024). All our methods outperformed other methods in terms of the three metrics on AIDS, PROTEINS, and REDDIT-MULTI-5K significantly. On ENZYMES, our MFD performs best. The differences between MMFD, MMFD$_{\text{LR}}$, and MMFD$_{\text{LR}}$-KM are tiny but their per-

| Method | AIDS ($N = 2000$) | | | PROTEINS ($N = 1113$) | | | ENZYMES ($N = 600$) | | | REDDIT-MULTI-5K ($N = 4999$) | | |
|---|---|---|---|---|---|---|---|---|---|---|---|---|
| | ACC | NMI | ARI | ACC | NMI | ARI | ACC | NMI | ARI | ACC | NMI | ARI |
| SP kernel | 79.49±0.84 | 0.39±0.62 | -0.71±1.13 | 64.42±0.00 | 6.03±0.00 | 5.87±0.00 | 22.00±0.00 | 2.57±0.00 | 1.69±0.00 | 20.02±0.00 | 0.05±0.00 | 0.00±0.00 |
| GK kernel | 79.95±0.00 | 0.04±0.00 | -0.07±0.00 | 59.61±0.22 | 0.24±0.18 | 0.10±0.19 | 17.07±0.13 | 0.80±0.25 | 0.00±0.00 | – | – | – |
| RW kernel | 79.90±0.00 | 0.09±0.00 | -0.15±0.00 | – | – | – | 17.00±0.00 | 0.66±0.00 | 0.25±0.00 | – | – | – |
| WL kernel | 78.50±0.00 | 1.17±0.00 | -2.09±0.00 | 60.38±0.00 | 1.55±0.00 | 0.81±0.00 | 21.00±0.00 | 3.09±0.00 | 1.48±0.00 | 20.00±0.00 | 0.00±0.00 | 0.00±0.00 |
| LT kernel | 79.95±0.00 | 0.04±0.00 | -0.07±0.00 | – | – | – | 17.00±0.09 | 0.42±0.11 | 0.00±0.00 | – | – | – |
| WL-OA kernel | 80.40±0.00 | 2.46±0.00 | 2.38±0.00 | 60.38±0.00 | 1.55±0.00 | 0.81±0.00 | 20.00±0.00 | 1.35±0.00 | 0.32±0.00 | – | – | – |
| InfoGraph+KM | 92.21±0.81 | 54.49±3.53 | 63.78±3.84 | 59.22±0.21 | 3.22±1.94 | 0.00±0.00 | 22.06±0.98 | 2.40±0.45 | 1.25±0.52 | 20.16±0.02 | 0.30±0.05 | 0.00±0.00 |
| InfoGraph+SC | 95.65±1.55 | 72.21±9.20 | 80.17±7.19 | 64.02±2.31 | 5.17±1.87 | 7.06±2.65 | 23.75±0.50 | 4.64±0.65 | 2.23±0.41 | 20.00±0.00 | 0.00±0.00 | 0.00±0.00 |
| GraphCL+KM | 90.40±1.06 | 46.56±4.31 | 55.29±5.28 | 59.47±0.01 | 0.37±0.31 | 0.00±0.00 | 21.50±0.22 | 1.55±0.12 | 0.90±0.09 | 20.32±0.00 | 0.56±0.00 | 0.00±0.00 |
| GraphCL+SC | 96.08±1.96 | 72.97±10.86 | 81.65±8.51 | 59.96±0.10 | 2.81±0.07 | 3.88±0.08 | 25.28±0.28 | 4.75±0.36 | 2.03±0.26 | 20.08±0.00 | 0.16±0.00 | 0.00±0.00 |
| JOAO+KM | 88.25±0.00 | 38.02±0.00 | 44.62±0.00 | 59.48±0.00 | 0.64±0.05 | -0.06±0.00 | 21.66±0.37 | 1.60±0.01 | 0.94±0.02 | 20.34±0.00 | 0.60±0.00 | 0.00±0.00 |
| JOAO+SC | 80.13±0.02 | 0.84±0.15 | 0.80±0.14 | 59.75±0.00 | 0.47±0.00 | 0.17±0.00 | 24.65±0.44 | 4.85±0.37 | 2.07±0.18 | 20.39±0.49 | 0.08±0.00 | 0.01±0.01 |
| GWF+KM | 96.43±1.71 | 74.48±9.15 | 84.71±7.02 | 66.87±2.36 | 9.07±1.21 | 11.43±3.19 | 28.55±0.20 | 6.02±0.55 | 3.16±0.20 | – | – | – |
| GWF+SC | 96.44±2.92 | 76.01±15.23 | 83.54±13.61 | 68.79±2.05 | 10.17±1.74 | 13.88±2.72 | 25.66±1.57 | 5.24±1.28 | 1.78±0.61 | – | – | – |
| GLCC | 79.02±0.62 | 4.18±2.01 | 5.05±2.13 | 60.65±2.69 | 2.08±1.43 | 4.16±2.28 | 19.89±1.09 | 2.42±0.18 | 0.19±0.12 | 23.50±0.48 | 6.57±3.56 | 4.00±0.80 |
| DCGLC | 96.77±0.33 | 73.51±2.30 | 85.74±1.45 | 68.89±2.04 | 10.90±1.35 | 14.32±2.88 | 28.43±1.28 | 6.57±0.20 | 3.78±0.47 | 33.24±2.34 | 8.81±2.28 | 7.16±1.67 |
| MMD | 50.10±0.00 | 0.00±0.00 | 0.03±0.00 | 52.56±0.00 | 0.08±0.00 | 0.14±0.00 | 22.90±1.14 | 1.79±0.28 | 0.47±0.22 | 33.31±0.91 | 17.68±0.17 | 11.20±0.84 |
| GWD | 88.30±0.00 | 49.73±0.00 | 56.45±0.00 | 68.82±0.00 | 12.42±0.00 | 12.37±0.00 | 23.08±0.37 | 3.91±0.69 | 0.41±0.11 | – | – | – |
| GED | 89.55±0.00 | 43.33±0.00 | 51.02±0.00 | 52.24±0.07 | 3.92±0.23 | -0.23±0.03 | 25.50±0.00 | 5.05±0.18 | 2.24±0.06 | – | – | – |
| **MMFD** | 98.80±0.00 | 88.37±0.00 | 94.49±0.00 | 72.60±0.00 | 14.18±0.00 | 19.67±0.00 | 23.68±1.31 | 5.99±0.78 | 1.96±0.33 | 35.97±0.48 | 18.12±0.61 | 13.74±1.97 |
| **MMFD$_{LR}$** | 98.80±0.00 | 88.37±0.00 | 94.49±0.00 | 72.49±0.13 | 13.98±0.25 | 19.49±0.23 | 23.62±0.83 | 6.55±0.42 | 2.13±0.22 | 36.54±0.56 | 18.31±0.74 | 13.99±2.18 |
| **MMFD$_{LR}$-KM** | 98.96±0.02 | 89.62±0.18 | 95.25±0.11 | 71.87±0.18 | 12.74±0.34 | 18.51±0.28 | 25.67±0.77 | 6.34±0.68 | 2.06±0.37 | 35.62±0.46 | 19.80±0.06 | 13.74±0.30 |
| **MFD** | 99.45±0.00 | 93.82±0.00 | 97.47±0.00 | 72.60±0.00 | 14.18±0.00 | 19.67±0.00 | 30.33±1.16 | 8.45±0.26 | 4.42±0.33 | 35.35±0.01 | 17.28±0.03 | 14.90±0.02 |
| **MFD-KD** | 99.02±0.00 | 90.01±0.34 | 95.51±0.18 | 72.39±0.30 | 14.06±0.40 | 19.24±0.57 | 26.25±1.44 | 7.54±0.69 | 2.21±0.60 | 34.86±0.22 | 19.70±0.29 | 14.46±0.36 |

Table 3: Clustering results (mean ± std of 10 trials) on the four moderate-size datasets. The best, second-best, and third-best results are highlighted in purple, red, and orange respectively. " – " denotes out-of-memory or taking more than 12 hours.

formances are lower than MFD and MFD-KD in most cases, since MFD can exploit more information of distributions.

Table 4 compares the time costs of four methods in computing the distance or similarity matrices on two datasets. We see that our MMFD$_{LR}$ is the fastest while GWD is extremely time-consuming.

| | AIDS | PROTEINS |
|---|---|---|
| Shortest-path kernel | 1.51 | 7.55 |
| WL-subtree kernel | 0.81 | 0.90 |
| Entropic GWD | 25544.26 | 4549.31 |
| **MMFD$_{LR}$** | **0.26** | **0.61** |

Table 4: Time costs (second) comparison.

Due to space limitations, the following results are presented in the appendices:

- **Distribution comparison (Appendix C.1)**
- **Effectiveness of node attributes (Appendix C.2)**
- **Hyperparameter analysis (Appendix C.3)**
- **More running time comparison (Appendix C.4)**
- **Visualization of graphs (Appendix C.5)**
- **More results on toy graphs (Appendix C.6)**
- **Experiments on larger datasets (Appendix C.7)**
- **Molecular property prediction (Appendix C.8)**

## 4. Conclusion

This work proposed MMFD and MFD as efficient and accurate distance measures between graphs. For large-scale clustering, we also proposed MMFD-KM and MFD-KD, which has a linear complexity with respect to the number of graphs in a dataset. We analyzed the properties of the proposed distance measures. We used synthetic graph data to show the effectiveness of MMFD and MFD intuitively. The experiments on real-world graph datasets showed that our methods are not only efficient but also accurate in clustering.

One limitation of the current work is that we haven't provided any numerical results of the variant of MFD proposed in Appendix A.3 because the optimization is too costly.

The Python code of the proposed algorithms is available at https://github.com/jicongfan/Graph-Minimum-Factorization-Distance.

## Acknowledgements

This work was supported by the National Natural Science Foundation of China under Grant No.62376236.

## Impact Statement

This paper presents work whose goal is to advance the field of Machine Learning. There are many potential societal consequences of our work, none which we feel must be specifically highlighted here.

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

# A. Beyond Mean Comparison: MFD

## A.1. Optimization of MFD

Based on Theorem 2.9, we can just let $\rho$ be a small positive constant such as $10^{-4}$. Figure 3 shows the optimization process of MFD with different $\rho$ on the $\tilde{G}_1$ and $\tilde{G}_2$ presented by Figure 2. In the experiments of real datasets, we set $T = 10$, which is sufficient.

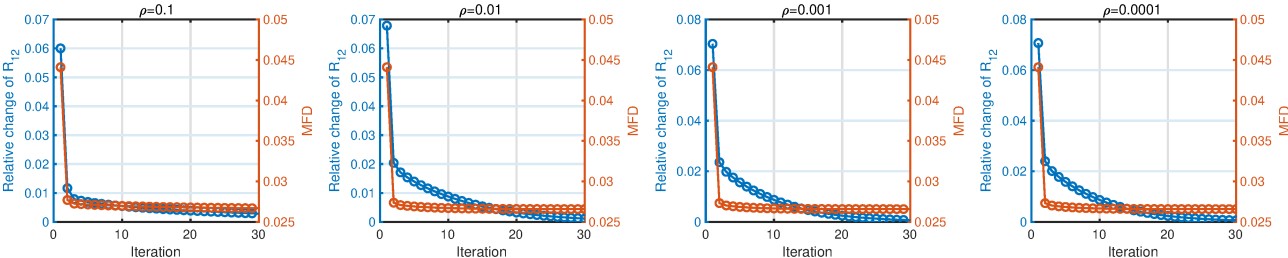

Figure 3: Optimization process of MFD $\tilde{G}_1$ and $\tilde{G}_2$ in Figure 2.

It is worth mentioning that we can perform down-sampling on $\mathbf{\Phi}_i$ to reduce the time cost of MFD. Suppose we randomly select $n_s$ columns of $\mathbf{\Phi}_i$ and then compute MFD, then the time complexity becomes $\mathcal{O}(n^2 \log(d) + d^2 n + T(d^3 + d n_s^2))$, where $n_s \ll n$. This can greatly improve the efficiency of computing the MFDs of $N$ graphs.

## A.2. MFD-KD for Large-Scale Clustering

Similar to MMFD, we can extend MFD to large-scale clustering by reducing the complexity dependency on $N^2$ to $N$. Specifically, we propose to solve

$$\underset{\bar{G}_1,\ldots,\bar{G}_K}{\text{minimize}} \sum_{l=1}^{K} \sum_{G_i \in \mathcal{G}_l} \text{MFD}^2(G_i, \bar{G}_l) \tag{28}$$

Instead of solving $\bar{G}_1, \ldots, \bar{G}_K$, we solve their features denoted as $\mathbf{\Psi}_1, \ldots, \mathbf{\Psi}_K \in \mathbb{R}^{M \times n}$. We have

$$\underset{\mathbf{\Psi}_1,\ldots,\mathbf{\Psi}_K}{\text{minimize}} \sum_{l=1}^{K} \sum_{G_i \in \mathcal{G}_l} \min_{\mathbf{R}_i \in \mathcal{R}} \left\| \frac{1}{n_i} \sum_{j=1}^{n_i} \zeta(\mathbf{R}_i \boldsymbol{\phi}_j^{(i)}) - \frac{1}{n} \sum_{j=1}^{n} \zeta(\boldsymbol{\psi}_j^{(l)}) \right\|^2 \triangleq \mathcal{L}(\mathbf{\Psi}_l, \mathbf{R}_i) \tag{29}$$

For convenience, we call this method MFD-KD, where 'KD' stands for 'K-distributions'. The optimization is a little bit similar to K-means clustering. First, we initialize $\mathbf{\Psi}_1^{(0)}, \ldots, \mathbf{\Psi}_K^{(0)}$. At iteration $t$, for each $G_i$, we find its label by solving

$$l_i^{(t)} = \underset{1 \leq l \leq K}{\text{argmin}} \ \min_{\mathbf{R}_i \in \mathcal{R}} \mathcal{L}(\mathbf{\Psi}_l^{(t-1)}, \mathbf{R}_i) \tag{30}$$

where the corresponding optimal $\mathbf{R}_i$ are denoted as $\mathbf{R}_i^{(t)}$. The optimization for (30) can be similarly solved by Algorithm 2. Base on $l_i^{(t)}$, $i = 1, \ldots, N$, we have the updated $\mathcal{G}_1, \ldots, \mathcal{G}_K$. Next, we update each of $\mathbf{\Psi}_1, \ldots, \mathbf{\Psi}_K$ by

$$\mathbf{\Psi}_l^{(t)} = \underset{\mathbf{\Psi}_l}{\text{argmin}} \sum_{G_i \in \mathcal{G}_l} \mathcal{L}(\mathbf{\Psi}_j, \mathbf{R}_i^{(t)}) \tag{31}$$

This can be solved by any gradient-based methods. We alternate between (30) and (31) until some convergence condition is reached.

Specifically, (31) is equivalent to

$$\underset{\mathbf{\Psi}_l}{\text{minimize}} \sum_{G_i \in \mathcal{G}_l} \frac{1}{n^2} \sum_{u,v} \exp(-\beta \|\boldsymbol{\psi}_u^{(l)} - \boldsymbol{\psi}_v^{(l)}\|^2) - \frac{2}{n_i n} \sum_{u,v} \exp(-\beta \|\boldsymbol{\phi}_u^{(i)} - \mathbf{R}_i^{(t)} \boldsymbol{\psi}_v^{(l)}\|^2) \tag{32}$$

Let $\mathcal{L}$ be the objective function in (32), we have the gradient

$$\nabla_{\boldsymbol{\Psi}_l}\mathcal{L} = \sum_{G_i \in \mathcal{G}_l} \frac{4\beta}{n^2} \boldsymbol{\Psi}_l(\mathbf{Q}_l - \text{diag}(\mathbf{1}^\top \mathbf{Q}_l)) - \frac{4\beta}{n_i n} \mathbf{R}_i^{(t)^\top}(\boldsymbol{\Phi}_i \mathbf{G}_l - \mathbf{R}_i^{(t)} \boldsymbol{\Psi}_l \text{diag}(\mathbf{1}^\top \mathbf{G}_l))$$

$$= \boldsymbol{\Psi}_l \sum_{G_i \in \mathcal{G}_l} \left( \frac{4\beta}{n^2}(\mathbf{Q}_l - \text{diag}(\mathbf{1}^\top \mathbf{Q}_l)) + \frac{4\beta}{n_i n}\text{diag}(\mathbf{1}^\top \mathbf{G}_l) \right) - \sum_{G_i \in \mathcal{G}_l} \frac{4\beta}{n_i n} \mathbf{R}_i^{(t)^\top} \boldsymbol{\Phi}_i \mathbf{G}_l \tag{33}$$

where $\mathbf{Q}_l = [\exp(-\beta\|\boldsymbol{\psi}_u^{(l)} - \boldsymbol{\psi}_v^{(l)}\|^2)]$ and $\mathbf{G}_l = [\exp(-\beta\|\boldsymbol{\phi}_u^{(i)} - \mathbf{R}_i^{(t)}\boldsymbol{\psi}_v^{(l)}\|^2)]$. Then we perform the following gradient descent

$$\boldsymbol{\Psi}_l \longleftarrow \boldsymbol{\Psi}_l - \eta\nabla_{\boldsymbol{\Psi}_l}\mathcal{L} \tag{34}$$

where $\eta = c/\|\mathbf{H}\|_2$, $c \leq 1$ is some constant, and $\mathbf{H} = \sum_{G_i \in \mathcal{G}_l} \left( \frac{4\beta}{n^2}(\mathbf{Q}_l - \text{diag}(\mathbf{1}^\top \mathbf{Q}_l)) + \frac{4\beta}{n_i n}\text{diag}(\mathbf{1}^\top \mathbf{G}_l) \right)$. Note that here $\|\mathbf{H}\|_2$ is a rough approximation of the Lipschitz constant of the gradient $\nabla_{\boldsymbol{\Psi}_l}\mathcal{L}$. In our experiments, we set $c = 0.5$.

The optimization steps of MFD-KD is summarized into Algorithm 3.

---

**Algorithm 3** MFD-KD Clustering

---

**Input:** Graphs $G_1, G_2, \ldots, G_N$ with self-looped adjacency matrices $\mathbf{A}_1, \mathbf{A}_2, \ldots, \mathbf{A}_N$, $K$, $d$, $\beta$, $\rho$, $\varepsilon$, $T_1$, $T_2$, $T_3$.
1: **for** i=1:N **do**
2:     $\mathbf{A}_i \approx \mathbf{U}_i\mathbf{S}_i\mathbf{V}_i^\top$ (randomized SVD, rank-$d$).
3:     $\mathcal{A}_i^\phi = \bar{\mathbf{V}}_i\bar{\mathbf{S}}_i\bar{\mathbf{V}}_i^\top$.
4:     $\boldsymbol{\Phi}_i = \bar{\mathbf{S}}_i^{1/2}\bar{\mathbf{V}}_i^\top$.
5: **end for**
6: Initialization: $\boldsymbol{\Psi}_1^{(0)}, \ldots, \boldsymbol{\Psi}_K^{(0)} \in \mathbb{R}^{M \times n}$, $t = 0$.
7: **while** $t < T_1$ **do**
8:     $t \leftarrow t + 1$.
9:     **for** i=1:N **do**
10:         Solve (30) for each $j$ using Algorithm 2 with $T_2$ iterations.
11:     **end for**
12:     **for** l=1:K **do**
13:         Solve (31) using gradient descent (e.g., (34)) with $T_3$ iterations.
14:     **end for**
15: **end while**
**Output:** $\mathcal{G}_1, \mathcal{G}_2, \ldots, \mathcal{G}_K$

---

### A.3. A Variant of MFD based on EMD

In MFD, instead of MMD, we can also use the optimal transportation theory to measure the distance between two distributions. The corresponding distance is the Wasserstein distance or Earth Mover's Distance (EMD). Let's consider a cost matrix defined as

$$\mathbf{C}(\mathbf{R}_{12}) := \text{cost}(\boldsymbol{\Phi}_1, \mathbf{R}_{12}\boldsymbol{\Phi}_i) \tag{35}$$

where cost : $\mathbb{R}^{M \times n_1} \times \mathbb{R}^{M \times n_2} \to \mathbb{R}^{n_1 \times n_2}$. Now let $\xi$ be the EMD and define $\mathcal{P} := \{\mathbf{P} \in \mathbb{R}^{n_1 \times n_2} : P_{uv} \geq 0, \mathbf{P}^\top\mathbf{1}_{n_1} = \mathbf{1}_{n_2}/n_2, \mathbf{P}\mathbf{1}_{n_2} = \mathbf{1}_{n_1}/n_1\}$, we obtain the following distance between $G_1$ and $G_2$

$$\text{MFD}(G_1, G_2) = \min_{\mathbf{R}_{12} \in \mathcal{R}, \mathbf{P}_{12} \in \mathcal{P}} \langle \mathbf{C}(\mathbf{R}_{12}), \mathbf{P}_{12}\rangle$$

$$= \min_{\mathbf{R}_{12} \in \mathcal{R}, \mathbf{P}_{12} \in \mathcal{P}} \langle \text{cost}(\boldsymbol{\Phi}_1, \mathbf{R}_{12}\boldsymbol{\Phi}_i), \mathbf{P}_{12}\rangle \tag{36}$$

The optimization of this $\text{MFD}(G_1, G_2)$ is challenging. Since there are two blocks of parameters to optimize, we propose to use alternating minimization, i.e.,

$$\mathbf{P}_{12}^{(t)} = \underset{\mathbf{P}_{12} \in \mathcal{P}}{\text{argmin}} \left\langle \text{cost}(\boldsymbol{\Phi}_1, \mathbf{R}_{12}^{(t-1)}\boldsymbol{\Phi}_i), \mathbf{P}_{12}\right\rangle \tag{37}$$

$$\mathbf{R}_{12}^{(t)} = \underset{\mathbf{R}_{12} \in \mathcal{R}}{\operatorname{argmin}} \left\langle \operatorname{cost}(\mathbf{\Phi}_1, \mathbf{R}_{12}\mathbf{\Phi}_i), \mathbf{P}_{12}^{(t)} \right\rangle \tag{38}$$

Subproblem (37) can be solved by Sinkhorn iteration, while subproblem (38) can be solved by a method similar to that for (22). Both subproblems are time-consuming. The total time complexity of MFD with EMD is much higher than that of MFD with MMD. In this work, we do not provide more details about this method.

## B. Experimental Settings

### B.1. Dataset Description

Table 5 presents the basic information of the datasets used in this work. They were downloaded from `https://chrsmrrs.github.io/datasets/`. All experiments were conducted on a computer with Intel Core i9-12900K and RAM 64GB.

| Name | Graphs | Average nodes | Average edges | Node Attributes | Classes | Domain |
|------|--------|---------------|---------------|-----------------|---------|--------|
| AIDS | 2,000 | 15.69 | 16.20 | 4 | 2 | Molecule |
| PROTEINS | 1,113 | 39.06 | 72.82 | 29 | 2 | Biology |
| ENZYMES | 600 | 32.63 | 62.14 | 18 | 6 | Biology |
| REDDIT-MULTI-5K | 4,999 | 508.52 | 594.87 | - | 5 | Social networks |
| DBLP-v1 | 19,456 | 10.48 | 19.65 | - | 2 | Social networks |
| REDDIT-MULTI-12K | 11,929 | 391.41 | 456.8 | - | 11 | Social networks |

Table 5: Statistics of the graph datasets used in this work.

### B.2. Hyperparameter Settings

Since we just used the results of the graph kernels, graph representation learning, GLCC, and DCGLC reported in the paper of (Cai et al., 2024), the corresponding settings of the hyperparameters or neural network architectures can be found in (Cai et al., 2024). We implement the entropic-regularized GWD using the function of POT `https://pythonot.github.io/index.html`, where the loss function is the square loss, the regularization hyperparameter $\epsilon$ is chosen from $\{0.01, 0.1, 1\}$, the maximum iteration is 100, and the iteration tolerance is 0.01. We implement GED using the function of graphkit-learn `https://graphkit-learn.readthedocs.io/en/master/index.html`, where the edit costs of edge and node insertions and deletions are all set to 1.

In our MMFD$_{\text{LR}}$ and MMFD$_{\text{LR}}$-KM, we set $d = 20$ on all datasets. In MFD and MFD-KD, we use a different rank $d = 30$ because we do not compare the full rank with low rank for simplicity. For spectral clustering, the $\gamma$ in the Gaussian kernel is set as $1/(cu)^2$, where $u$ denotes the average MMDF (or MMD, GWD, and GED) between all graphs and $c = 5$ for all datasets. On ENZYMES, we use the node attributes in MMFD, MMFD$_{\text{LR}}$, and MMFD$_{\text{LR}}$-KM, MFD, and MFD-KD, where the graphs based on the attributes are constructed by Gaussian kernel. In MFD and MFD-KD, the hyperparameter $\beta$ is chosen from $\{0.01, 0.1, 1\}$. For MFD (Algorithm 2), we set $T = 10$. For MFD-KD (Algorithm 3), we set $T_2 = T_3 = 5$, on all datasets, set $T_1 = 10$ on large datasets (i.e., the two REDDIT datasets), and set $T_1 = 20$ on other datasets. Particularly, on the REDDIT-MULTI-5K dataset, for MFD, we let $d = 5$ and use down-sampling on $\mathbf{\Phi}_i$ with a sampling rate 0.2 to improve the efficiency, which leads to a total time cost of 5.5 hours.

## C. More Results

### C.1. Effectiveness of Distribution Comparison to Distinguish Graphs

We randomly generate $\mathbf{Z}_1, \mathbf{Z}_2 \in \mathbb{R}^{10 \times 100}$ from the same Gaussian distribution with zero mean and unit variance and generate $\mathbf{Z}_3 \in \mathbb{R}^{10 \times 100}$ from a uniform distribution with zero mean and unit variance. From $\mathbf{Z}_1, \mathbf{Z}_2, \mathbf{Z}_3$, we construct three graphs using the kernel function $k(\mathbf{z}_u, \mathbf{z}_v) = \mathbb{1}(\exp(-0.1\|\mathbf{z}_u - \mathbf{z}_v\|^2) > 0.5)$ respectively. We then calculate the MMFDs between the three graphs. For instance, in one experiment, we have MMFD($G_1, G_2$)=0.0038 and MMFD($G_1, G_3$)=0.0276, meaning that $G_1$ is more similar to $G_2$ than to $G_3$. We repeat the experiment 100 times and conduct t-test on the values of MMFD($G_1, G_2$) and MMFD($G_1, G_3$). The $p$-value is less than $10^{-6}$, meaning that the difference is significant. When we also generate $\mathbf{Z}_3$ from the Gaussian distribution and repeat the experiment 100 times, the $p$-value of t-test between MMFD($G_1, G_2$) and MMFD($G_1, G_3$) is 0.6864, meaning that the difference is not significant. These results demonstrated

that comparing distributions is an effective proxy of comparing graphs.

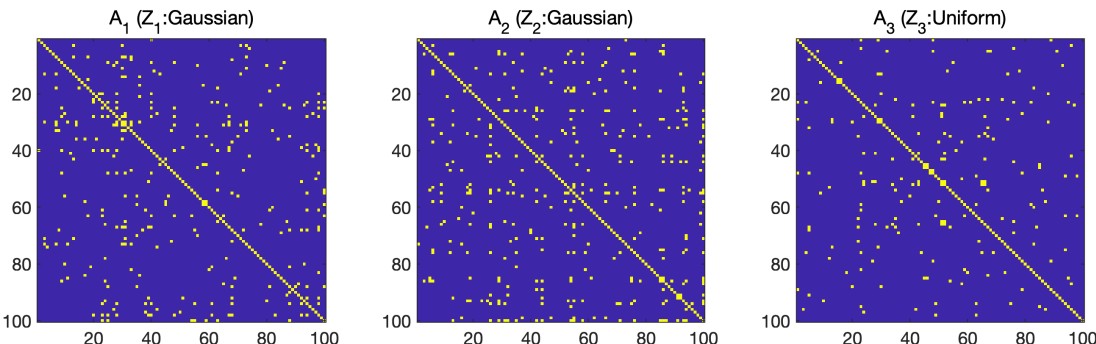

Figure 4: Visualization of the adjacency matrices of three graphs $G_1, G_2, G_3$ constructed from two 10-D Gaussian distributions (zero mean and unit variance) and one 10-D uniform distribution (unit variance) respectively. The kernel function is $k(\mathbf{z}_u, \mathbf{z}_v) = \mathbb{1}(\exp(-0.1\|\mathbf{z}_u - \mathbf{z}_v\|^2) > 0.5)$. We have MMFD$(G_1, G_2)$=0.0038 and MMFD$(G_1, G_3)$=0.0276.

### C.2. Effectiveness of Node Attributes

AIDS, PROTEINS, and ENZYMES have inherent node attributes. Therefore, we use them to evaluate the effectiveness of incorporating node attributes into our proposed methods. For simplicity, we take MMFD$_{LR}$-KM as an example. The results are reported in Table 6. We see that using node attributes can improve the clustering performance, and the improvement on PROTEINS is most significant.

|  | AIDS | | | PROTEINS | | | ENZYMES | | |
|---|---|---|---|---|---|---|---|---|---|
|  | ACC | NMI | ARI | ACC | NMI | ARI | ACC | NMI | ARI |
| without node attributes | 98.96±0.02 | 89.62±0.18 | 95.25±0.11 | 71.87±0.18 | 12.74±0.34 | 18.51±0.28 | 23.27±0.28 | 3.34±0.37 | 0.82±0.09 |
| with node attributes | **99.10±0.00** | **90.68±0.00** | **95.87±0.00** | **75.19±0.05** | **18.17±0.10** | **24.81±0.11** | **25.67±0.77** | **6.34±0.68** | **2.06±0.37** |

Table 6: Clustering performance (mean ± std of 10 trials) of MMFD$_{LR}$-KM with or without using node attributes.

### C.3. Hyperparameter Analysis

In our MMFD, there is no hyperparameter. In our MMFD$_{LR}$ and MMFD$_{LR}$-KM, the only hyperparameter is the rank $d$. Here we take the datasets PROTEINS and REDDIT-MULTI-5K as examples to show the impact of $d$ on the clustering performance of MMFD$_{LR}$. The average results of 10 trials are shown in Table 7. On PROTEINS, since the average number of nodes is quite small, we have chosen $d$ from 1 to 40 or used full-rank. We see that when $d$ decreases, the performance in terms of the three metrics becomes worse. On REDDIT-MULTI-5K, rank-10 performs the best. In general, the performance of MMFD$_{LR}$ is not very sensitive to $r$. Even when the rank is 1, our MMFD$_{LR}$ still outperforms the competitors in Table 3.

| Dataset | Metrics | rank | | | | | | | |
|---|---|---|---|---|---|---|---|---|---|
|  |  | full-rank | rank-40 | rank-30 | rank-20 | rank-10 | rank-5 | rank-3 | rank-1 |
| PROTEINS | ACC | **72.60±0.00** | **72.60±0.00** | **72.60±0.00** | 72.49±0.13 | 72.31±0.14 | 72.08±0.04 | 71.83±0.04 | 72.13±0.13 |
|  | NMI | **14.18±0.00** | **14.18±0.00** | **14.18±0.00** | 13.98±0.25 | 13.62±0.29 | 13.14±0.10 | 12.55±0.07 | 13.42±0.14 |
|  | ARI | **19.67±0.00** | **19.67±0.00** | **19.67±0.00** | 19.49±0.23 | 19.22±0.24 | 18.83±0.06 | 18.49±0.08 | 18.83±0.26 |
|  |  | full-rank | rank-200 | rank-100 | rank-50 | rank-20 | rank-10 | rank-3 | rank-1 |
| REDDIT-MULTI-5K | ACC | 35.97±0.48 | 36.13±0.64 | 36.33±0.56 | 36.12±0.91 | 36.18±0.80 | **36.85±0.81** | 36.60±0.82 | 35.77±0.59 |
|  | NMI | 18.12±0.61 | 18.09±0.62 | 18.45±0.59 | 18.06±0.61 | 18.62±0.65 | **18.83±0.20** | 17.83±0.33 | 17.01±0.32 |
|  | ARI | 13.74±1.97 | 13.65±2.11 | 14.57±1.95 | 12.46±2.12 | 13.87±1.96 | **15.08±2.21** | 13.56±0.19 | 12.60±1.32 |

Table 7: Impact of the rank $d$ on the clustering performance of MMFD$_{LR}$. The best result in each case is highlighted in bold.

## C.4. Runing Time Comparison

First, we study the time cost of our method and a few baselines in comparing two graphs. We generate random graphs with the number of nodes $n$ increasing from 100 to 30000, where the number of edges of each graph is about $0.1n^2$. We compare the time costs (second) of MMFD$_{\text{LR}}$, Shortest-path kernel, WL subtree kernel, and Gromov-Wasserstein distance with entropic regularizer. In our MMFD$_{\text{LR}}$, we set $d = \log(n)$. The results are reported in Table 8. We see that the Shortest-path kernel is most time-consuming while our MMFD$_{\text{LR}}$ is always the most efficient one.

| $n$ | 100 | 1000 | 2000 | 3000 | 4000 | 5000 | 6000 | 7000 | 8000 | 9000 | 10000 | 20000 | 30000 |
|---|---|---|---|---|---|---|---|---|---|---|---|---|---|
| Shortest-path kernel | 0.035 | 4.853 | 25.8 | 77.8 | 178.7 | 317.9 | 524.8 | 779.3 | 1125.0 | 1520.3 | 2022.4 | – | – |
| WL subtree kernel | 0.009 | 0.393 | 1.5 | 3.4 | 6.6 | 10.5 | 15.3 | 21.1 | 27.4 | 34.9 | 43.4 | 182.6 | 449.7 |
| Gromov-Wasserstein | 0.021 | 0.367 | 0.5 | 1.2 | 2.3 | 3.9 | 6.4 | 9.0 | 12.8 | 17.1 | 21.9 | 169.1 | 579.7 |
| MMFD$_{\text{LR}}$ | **0.003** | **0.095** | **0.3** | **0.5** | **1.1** | **1.4** | **2.1** | **2.6** | **3.7** | **4.4** | **6.4** | **29.9** | **78.8** |

Table 8: Time cost (second) comparison on two random graphs. The number of nodes $n$ increases from 100 to 30000.

Second, we compare the time costs of our method with the three baselines in computing the distance or similarity matrices of a set of graphs. We consider AID, PROTEINS, and ENZYMES only because the Gromov-Wasserstein distance with entropic regularizer is not scalable to larger graphs such as the REDDIT datasets. The results are reported in Table 9. Since the graphs in the datasets are small, the Shortest-path kernel has a similar time cost as the WL-subtree kernel and our MMFD$_{\text{LR}}$. The Gromov-Wasserstein distance, though having used the entropic regularizer, is time-consuming.

| | AIDS (N=2000) | PROTEINS (N=1113) | ENZYMES (N=600) |
|---|---|---|---|
| Shortest-path kernel | 1.51 | 7.55 | 1.34 |
| WL subtree kernel | 0.81 | 0.90 | 0.38 |
| Gromov-Wasserstein | 25544.26 | 4549.31 | 1600.87 |
| MMFD$_{\text{LR}}$ | **0.26** | **0.61** | **0.14** |

Table 9: Time cost (second) comparison on three moderate-size datasets.

## C.5. Graph Visualization

According to the final format of MMFD, i.e. (12), we can visualize graphs in one-dimensional space directly using $\frac{1}{n_i}\sqrt{\sum_{uv}[\mathcal{A}_i^\phi]_{uv}}, i = 1, \ldots, N$. Regarding MFD, we can apply t-SNE (Van der Maaten & Hinton, 2008) to the obtained distance matrix between graphs to visualize graphs in 2-D space. As shown in Figure 5, we compare MMFD and MFD with InfoGraph and DCGLC by visualizing graphs of AIDS in 2-D space. For InfoGraph and DCGLC, we just use the t-SNE plots reported in (Cai et al., 2024). We see that among the four methods, MFD has the best separation between the two clusters, which is consistent with its high ACC, NMI, and ARI.

## C.6. More Results on Toy Graphs

### C.6.1. MMFD V.S. OTHER MEASURES ON TOY GRAPHS

We compare MMFD with GED, GWD (without entropic regularizer), and WL-subtree kernel on the seven synthetic graphs used in Table 2. We implement the WL-subtree kernel using the function of GraKel https://ysig.github.io/GraKeL/0.1a8/, where the number of iterations is 3 and the node labels are all one. To be consistent with other methods, we regard $1 - k(G_i, G_j)$ as the distance between $G_i$ and $G_j$. The results are shown in Table 10. We have the following observations.

- Regarding GED, for each of $G_2, G_4, G_7$, it cannot provide a unique closest graph. To each graph, there are always two graphs with the same distances.

- Regarding GWD, the distances of $G_6$ and $G_7$ to $G_4$ are the same. The closest graph to $G_7$ is identified as $G_4$, which is not reliable.

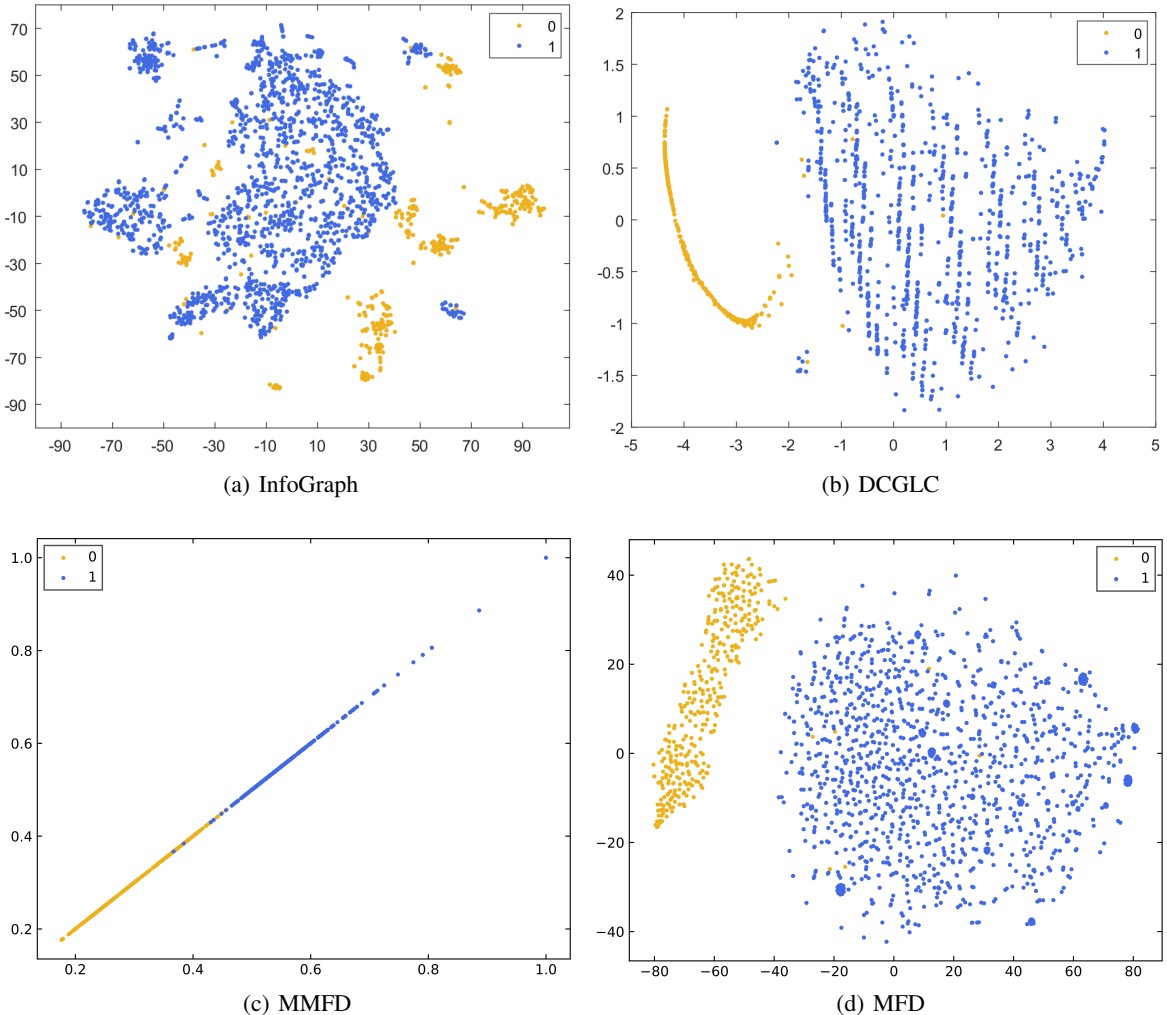

Figure 5: Visualization of graphs in AIDS. Plots (a), (b), and (d) are given by t-SNE, while (c) is the direct visualization of 1-D embedding (expanded to 2-D) of MMFD. The number of classes in the dataset is two.

- Regarding the WL-subtree kernel, it does not distinguish between $G_1, G_2, G_3$. The distance between $G_7$ and $G_4$ is much less than the distance between $G_7$ and $G_5$ (or $G_6$), which is not reliable.

### C.6.2. MFD v.s. MMFD on Toy Graphs

We compare MFD with MMFD on four synthetic graphs in Table 11. Intuitively, $G_1$ and $G_2$ are more similar to each other than to $G_3$ and $G_4$, and vice versa. MMFD with full rank failed to identify the most similar graphs to $G_2$ and $G_3$, while MFD with rank one succeeded. In contrast, both MFD with full rank and MDF with rank one identified all similarities correctly. These results demonstrated the superiority of MDF over MMFD. The fundamental reason is that MFD can exploit more information of graphs in comparison to MMFD.

### C.7. Clustering Results on Larger Datasets

We compare our methods with only three competitors, including WL-subtree kernel, InfoGraph+KM, and DCGLC, on two large datasets DBLP-v1 and REDDIT-MULTI-12K because other competitors are too time-consuming and often outperformed by the three methods. Nevertheless, we run the three methods only once because they are still costly, while our methods are run 10 times because of their high efficiency. The results are reported in Table 12. Our methods outperformed

(a) MMFD

| | $G_1$ | $G_2$ | $G_3$ | $G_4$ | $G_5$ | $G_6$ | $G_7$ |
|---|---|---|---|---|---|---|---|
| $G_1$ | – | **0.0914** | 0.1589 | 0.2097 | 0.2528 | 0.2505 | 0.2505 |
| $G_2$ | 0.0914 | – | **0.0675** | 0.1182 | 0.1614 | 0.1590 | 0.1591 |
| $G_3$ | 0.1589 | 0.0675 | – | **0.0507** | 0.0939 | 0.0915 | 0.0916 |
| $G_4$ | 0.2097 | 0.1182 | 0.0507 | – | 0.0432 | **0.0408** | 0.0409 |
| $G_5$ | 0.2528 | 0.1614 | 0.0939 | 0.0432 | – | 0.0024 | **0.0023** |
| $G_6$ | 0.2505 | 0.1590 | 0.0915 | 0.0408 | 0.0024 | – | **0.0001** |
| $G_7$ | 0.2505 | 0.1591 | 0.0916 | 0.0409 | 0.0023 | **0.0001** | – |

(b) GED

| | $G_1$ | $G_2$ | $G_3$ | $G_4$ | $G_5$ | $G_6$ | $G_7$ |
|---|---|---|---|---|---|---|---|
| $G_1$ | – | **4** | 6 | 10 | 10 | 12 | 12 |
| $G_2$ | **4** | – | **4** | 6 | 8 | 12 | 10 |
| $G_3$ | 6 | 4 | – | **2** | 4 | 10 | 8 |
| $G_4$ | 10 | 6 | **2** | – | **2** | 8 | 6 |
| $G_5$ | 10 | 8 | 4 | **2** | – | 6 | 4 |
| $G_6$ | 12 | 12 | 10 | 8 | 6 | – | **4** |
| $G_7$ | 12 | 10 | 8 | 6 | **4** | **4** | – |

(c) GWD

| | $G_1$ | $G_2$ | $G_3$ | $G_4$ | $G_5$ | $G_6$ | $G_7$ |
|---|---|---|---|---|---|---|---|
| $G_1$ | – | 0.4500 | **0.2500** | 0.3214 | 0.5625 | 0.5625 | 0.5625 |
| $G_2$ | 0.4500 | – | **0.1444** | 0.1845 | 0.2275 | 0.2275 | 0.2275 |
| $G_3$ | 0.2500 | 0.1444 | – | **0.1349** | 0.1806 | 0.1806 | 0.1528 |
| $G_4$ | 0.3124 | 0.1845 | 0.1349 | – | 0.1658 | **0.1071** | **0.1071** |
| $G_5$ | 0.5625 | 0.2275 | 0.1806 | 0.1071 | – | **0.0625** | 0.1250 |
| $G_6$ | 0.5625 | 0.2275 | 0.1806 | 0.1071 | **0.0625** | – | 0.1250 |
| $G_7$ | 0.5625 | 0.2275 | 0.1528 | **0.1071** | 0.1875 | 0.1250 | – |

(d) WL-subtree kernel

| | $G_1$ | $G_2$ | $G_3$ | $G_4$ | $G_5$ | $G_6$ | $G_7$ |
|---|---|---|---|---|---|---|---|
| $G_1$ | – | **0** | **0** | 0.2079 | 0.3610 | 0.4855 | 0.3965 |
| $G_2$ | **0** | – | **0** | 0.2079 | 0.3610 | 0.4855 | 0.3965 |
| $G_3$ | **0** | **0** | – | 0.2079 | 0.3610 | 0.4855 | 0.3965 |
| $G_4$ | 0.2079 | 0.2079 | 0.2079 | – | 0.1323 | 0.1849 | **0.1083** |
| $G_5$ | 0.3610 | 0.3610 | 0.3610 | **0.1323** | – | 0.2172 | 0.1693 |
| $G_6$ | 0.4855 | 0.4855 | 0.4855 | 0.18749 | 0.2172 | – | **0.1401** |
| $G_7$ | 0.3965 | 0.3965 | 0.3965 | **0.1083** | 0.1694 | 0.1401 | – |

Table 10: Comparison between MMFD and other measures on seven synthetic graphs ($G_1, G_2, \ldots, G_7$ from left to right). In each row of each table, for the corresponding graph, the cell of the value of the numerically closest graph is highlighted in red.

other methods in most cases.

## C.8. Chemical Molecular Property Prediction

We combine MMFD with support vector regression and call this combination MMFD+SVR. We apply it to the QM9 dataset. Since kernel SVR is not scalable to very large datasets, we only use 25000 graphs for training and 5000 graphs for testing. Some results of MAE are in Table 13, where the classic MPNN and enn-s2s (Gilmer et al., 2017) are compared. We see MMFD+SVR works well. It still has a lot of room for improvement, e.g., by using attributes more effectively, using the extension MFD, or using more training data.

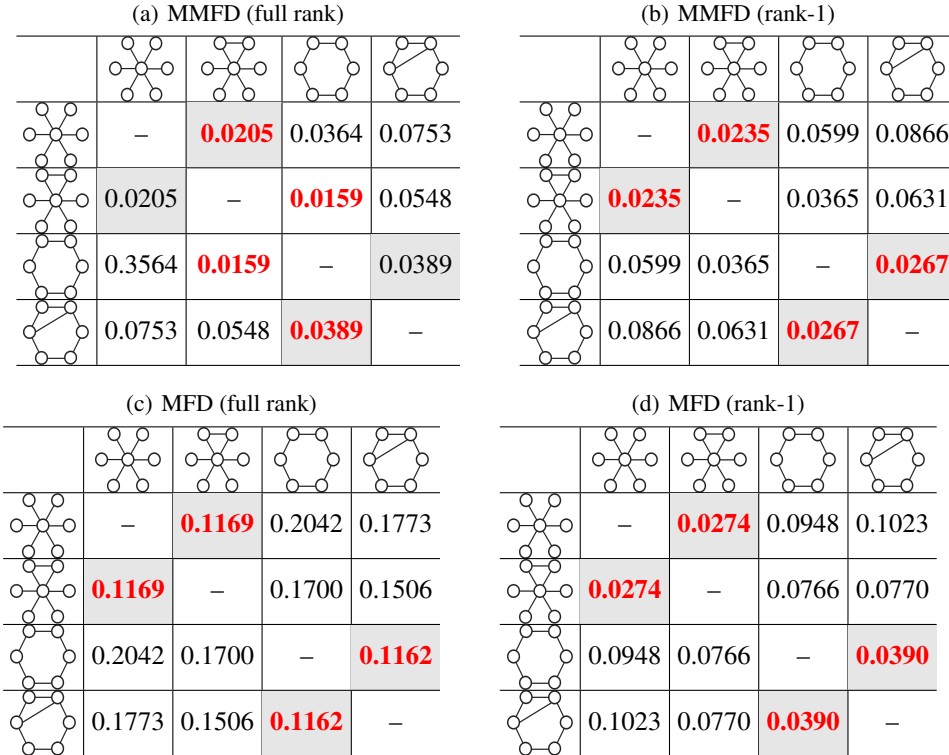

Table 11: Comparison between MMFD and MFD on four synthetic graphs ($G_1, G_2, G_3, G_4$ from left to right). In each row of each table, for the corresponding graph, the cell of the expected closest graph is marked in gray and the value of the numerically closest graph is highlighted in red.

| Dataset | DBLP-v1 ($N = 19456$) | | | REDDIT-MULTI-12K ($N = 11929$) | | |
|---|---|---|---|---|---|---|
| Metric | ACC | NMI | ARI | ACC | NMI | ARI |
| WL-subtree kernel | 55.04 | 5.41 | 0.99 | 25.37 | 10.98 | 6.96 |
| InfoGraph+KM | 59.47 | 11.45 | 3.56 | 22.20 | 8.86 | 2.70 |
| DCGLC | 75.73 | 22.07 | 26.49 | 22.03 | 3.25 | 0.23 |
| MMFD | 76.13±0.00 | 20.77±0.00 | 27.31±0.00 | 20.64±0.65 | 10.04±0.46 | 4.58±0.76 |
| MMFD-KM | 76.12±0.00 | 20.76±0.00 | 27.29±0.00 | 21.01±0.18 | 12.03±0.10 | 5.04±0.21 |
| MFD-KD | 78.95±0.00 | 25.84±0.50 | 33.53±0.73 | 22.83±0.48 | 15.89±0.43 | 7.91±0.12 |

Table 12: Clustering performance on DBLP-v1 and REDDIT-MULTI-12K. The best, second-best, and third-best results are highlighted in purple, red, and orange, respectively.

| Target | $\mu$ | $\alpha$ | HOMO | LUMO | gap | R2 | ZPVE | U0 |
|---|---|---|---|---|---|---|---|---|
| MPNN | 1.22 | 1.55 | 1.17 | 1.08 | 1.70 | 3.99 | 2.52 | 3.02 |
| enn-s2s | **0.30** | 0.92 | 0.99 | 0.87 | 1.60 | **0.15** | 1.27 | 0.45 |
| MMFD+SVR | 0.64 | **0.34** | **0.64** | **0.43** | **0.43** | 0.54 | **0.07** | **0.41** |

Table 13: Molecular property prediction on QM9 dataset.

# D. Proof for Theorems

## D.1. Proof for Theorme 2.2

*Proof.* It is obvious that $\text{MMFD}(G_1, G_2) \geq 0$, $\text{MMFD}(G_1, G_2) = \text{MMFD}(G_2, G_1)$, and $\text{MMFD}(G_1, G_1) = 0$ always hold. So we only need to prove the triangle inequality. Let $\mathbf{R}_{ij}^* = \arg\min_{\mathbf{R}_{ij} \in \mathcal{R}} \|\boldsymbol{\mu}_i - \mathbf{R}_{ij}\boldsymbol{\mu}_j\|$ and consider an arbitrary matrix $\mathbf{W} \in \mathbb{R}^{M \times M}$. Then based on the definition of MMFD, we have

$$
\begin{aligned}
&\text{MMFD}(G_1, G_2) \\
&= \|\boldsymbol{\mu}_1 - \mathbf{R}_{12}^* \boldsymbol{\mu}_2\| \\
&\leq \|\boldsymbol{\mu}_1 - \mathbf{W}\boldsymbol{\mu}_2\| \\
&= \|\boldsymbol{\mu}_1 - \mathbf{R}_{13}^* \boldsymbol{\mu}_3 + \mathbf{R}_{13}^* \boldsymbol{\mu}_3 - \mathbf{W}\boldsymbol{\mu}_2\| \\
&\leq \|\boldsymbol{\mu}_1 - \mathbf{R}_{13}^* \boldsymbol{\mu}_3\| + \|\mathbf{R}_{13}^* \boldsymbol{\mu}_3 - \mathbf{W}\boldsymbol{\mu}_2\| \\
&= \text{MMFD}(G_1, G_3) + \|\boldsymbol{\mu}_2 - \mathbf{W}^\top \mathbf{R}_{13}^* \boldsymbol{\mu}_3\|
\end{aligned}
\tag{39}
$$

Let $\mathbf{W}^\top \mathbf{R}_{13}^* = \mathbf{R}_{23}^*$, which has a solution $\hat{\mathbf{W}} = (\mathbf{R}_{23}^* (\mathbf{R}_{13}^*)^{-1})^\top = \mathbf{R}_{13}^* (\mathbf{R}_{23}^*)^\top$. We also have $\hat{\mathbf{W}}^\top \hat{\mathbf{W}} = \mathbf{R}_{23}^* (\mathbf{R}_{13}^*)^\top \mathbf{R}_{13}^* (\mathbf{R}_{23}^*)^\top = \mathbf{I}_M$, which means $\hat{\mathbf{W}} \in \mathcal{R}$. Now letting the $\mathbf{W}$ in (39) be $\hat{\mathbf{W}}$, we obtain

$$
\text{MMFD}(G_1, G_2) \leq \text{MMFD}(G_1, G_3) + \text{MMFD}(G_2, G_3)
\tag{40}
$$

$\square$

## D.2. Proof for Proposition 2.3

*Proof.* A complete graph is a graph in which each vertex is connected to every other vertex. Therefore, for two complete graphs, their self-looped adjacency matrices are $\mathbf{A}_i = [1]_{n_i \times n_i}$, $i = 1, 2$. $\mathbf{A}_i$ are PSD and rank-1. We have $\boldsymbol{\Phi}_i = s \cdot [1]_{1 \times n_i}$, where $s$ is $-1$ or $1$. That means $\boldsymbol{\mu}_i = -1$ or $+1$. The rotation matrix $\mathbf{R}_{12}$ is now just a scalar equal to $-1$ or $+1$. Therefore, $\min_{\mathbf{R}_{12} \in \mathcal{R}} \|\boldsymbol{\mu}_1 - \mathbf{R}_{12}\boldsymbol{\mu}_2\| \equiv 0$, for any $(n_1, n_2)$. $\square$

## D.3. Proof for Theorem 2.4

*Proof.* Let $\mathbf{1}_{n_i}$ be the $n_i$-dimensional vector with all entries 1 and $\mathbf{H}_{ij} = \mathbf{1}_{n_i} \mathbf{1}_{n_j}^\top$. It is easy to show that

$$
\begin{aligned}
&\text{MMFD}(G_1, G_2) \\
&= \min_{\mathbf{R}_{12} \in \mathcal{R}} \Big( \frac{1}{n_1^2} \mathbf{1}_{n_1}^\top \boldsymbol{\Phi}_1^\top \boldsymbol{\Phi}_1 \mathbf{1}_{n_1} + \frac{1}{n_2^2} \mathbf{1}_{n_2}^\top \boldsymbol{\Phi}_2^\top \mathbf{R}_{12}^\top \mathbf{R}_{12} \boldsymbol{\Phi}_2 \mathbf{1}_{n_2} \\
&\qquad\qquad - \frac{2}{n_1 n_2} \mathbf{1}_{n_1}^\top \boldsymbol{\Phi}_1^\top \mathbf{R}_{12} \boldsymbol{\Phi}_2 \mathbf{1}_{n_2} \Big)^{1/2} \\
&= \min_{\mathbf{R}_{12} \in \mathcal{R}} \Big( \frac{1}{n_1^2} \langle \boldsymbol{\mathcal{A}}_1^\phi, \mathbf{H}_{11} \rangle + \frac{1}{n_2^2} \langle \boldsymbol{\mathcal{A}}_2^\phi, \mathbf{H}_{22} \rangle - \frac{2}{n_1 n_2} \langle \mathbf{R}_{12}, \boldsymbol{\Phi}_1 \mathbf{H}_{12} \boldsymbol{\Phi}_2^\top \rangle \Big)^{1/2}
\end{aligned}
\tag{41}
$$

This indicates that for MMFD, we need to solve

$$
\underset{\mathbf{R}_{12} \in \mathcal{R}}{\text{maximize}} \ \langle \mathbf{R}_{12}, \mathbf{Q} \rangle
\tag{42}
$$

where $\mathbf{Q} = \boldsymbol{\Phi}_1 \mathbf{H}_{12} \boldsymbol{\Phi}_2^\top$. Problem (42) (also (10)) is equivalent to the well-known orthogonal Procrustes problem (Schönemann, 1966) and the optimal solution is

$$
\mathbf{R}_{12}^* = \mathbf{U_Q} \mathbf{V_Q}^\top
\tag{43}
$$

where $\mathbf{U_Q}, \mathbf{V_Q}$ are from the SVD of $\mathbf{Q}$, i.e., $\mathbf{Q} = \mathbf{U_Q}\mathrm{diag}(\sigma_1(\mathbf{Q}), \ldots, \sigma_n(\mathbf{Q}))\mathbf{V_Q}^\top$ with $\sigma_j(\mathbf{Q})$ denoting the $j$-th largest singular value of $\mathbf{Q}$. Since $\langle \mathbf{R}_{12}^*, \mathbf{Q} \rangle = \sum_j \sigma_j(\mathbf{Q})$, it follows from (41) that

$$
\begin{aligned}
&\mathrm{MMFD}(G_1, G_2) \\
&= \sqrt{\frac{1}{n_1^2} \sum_{uv} [\mathcal{A}_1^\phi]_{uv} + \frac{1}{n_2^2} \sum_{uv} [\mathcal{A}_2^\phi]_{uv} - \frac{2}{n_1 n_2} \sum_j \sigma_j(\mathbf{\Phi}_1 \mathbf{H}_{12} \mathbf{\Phi}_2^\top)} \\
&= \sqrt{\frac{1}{n_1^2} \sum_{uv} [\mathcal{A}_1^\phi]_{uv} + \frac{1}{n_2^2} \sum_{uv} [\mathcal{A}_2^\phi]_{uv} - \frac{2}{n_1 n_2} \sqrt{\sum_{uv} [\mathcal{A}_1^\phi]_{uv} \sum_{uv} [\mathcal{A}_2^\phi]_{uv}}} \\
&= \left| \frac{1}{n_1} \sqrt{\sum_{uv} [\mathcal{A}_1^\phi]_{uv}} - \frac{1}{n_2} \sqrt{\sum_{uv} [\mathcal{A}_2^\phi]_{uv}} \right|
\end{aligned}
\tag{44}
$$

The second equality in (44) holds because $\mathbf{\Phi}_1 \mathbf{H} \mathbf{\Phi}_2^\top$ is a rank-one matrix due to $\mathbf{H}$, which made $\sum_j \sigma_j(\mathbf{\Phi}_1 \mathbf{H}_{12} \mathbf{\Phi}_2^\top) = \sigma_1(\mathbf{\Phi}_1 \mathbf{H}_{12} \mathbf{\Phi}_2^\top) + 0 = \|\mathbf{\Phi}_1 \mathbf{1}_{n_1}\| \|\mathbf{\Phi}_2 \mathbf{1}_{n_2}\| = \sqrt{\mathbf{1}_{n_1}^\top \mathbf{\Phi}_1^\top \mathbf{\Phi}_1 \mathbf{1}_{n_1}} \sqrt{\mathbf{1}_{n_1}^\top \mathbf{\Phi}_2^\top \mathbf{\Phi}_2 \mathbf{1}_{n_2}} = \sqrt{\left(\sum_{uv} [\mathcal{A}_1^\phi]_{uv}\right)\left(\sum_{uv} [\mathcal{A}_2^\phi]_{uv}\right)}.$ $\quad\square$

### D.4. Proof for Theorem 2.5

*Proof.* For convenience, we denote $\tilde{\mathbf{A}}_i := \mathbf{A}_i + \mathbf{\Delta}_i$, $i = 1, 2$. Given the SVDs $\mathbf{A}_i = \mathbf{U}_i \mathbf{S}_i \mathbf{V}_i^\top$ and $\tilde{\mathbf{A}}_i = \tilde{\mathbf{U}}_i \tilde{\mathbf{S}}_i \tilde{\mathbf{V}}_i^\top$, we have $\mathcal{A}_i^\phi = \mathbf{V}_i \mathbf{S}_i \mathbf{V}_i^\top$ and $\tilde{\mathcal{A}}_i^\phi = \tilde{\mathbf{V}}_i \tilde{\mathbf{S}}_i \tilde{\mathbf{V}}_i^\top$. Letting $\mathbf{E}_i = \tilde{\mathcal{A}}_i^\phi - \mathcal{A}_i^\phi$, we have

$$
\begin{aligned}
&\left| \mathrm{MMFD}(G_1', G_2') - \mathrm{MMFD}(G_1, G_2) \right| \\
&= \frac{1}{n} \left| \left| \sqrt{\sum_{uv} [\mathcal{A}_1^\phi + \mathbf{E}_1]_{uv}} - \sqrt{\sum_{uv} [\mathcal{A}_2^\phi + \mathbf{E}_2]_{uv}} \right| - \left| \sqrt{\sum_{uv} [\mathcal{A}_1^\phi]_{uv}} - \sqrt{\sum_{uv} [\mathcal{A}_2^\phi]_{uv}} \right| \right| \\
&\le \frac{1}{n} \left| \sqrt{\sum_{uv} [\mathcal{A}_1 + \mathbf{E}_1]_{uv}} - \sqrt{\sum_{uv} [\mathcal{A}_1^\phi]_{uv}} + \sqrt{\sum_{uv} [\mathcal{A}_2^\phi]_{uv}} - \sqrt{\sum_{uv} [\mathcal{A}_2^\phi + \mathbf{E}_1]_{uv}} \right| \\
&\le \frac{1}{n} \left| \sqrt{\sum_{uv} [\mathcal{A}_1^\phi + \mathbf{E}_1]_{uv}} - \sqrt{\sum_{uv} [\mathcal{A}_1^\phi]_{uv}} \right| + \left| \sqrt{\sum_{uv} [\mathcal{A}_2^\phi]_{uv}} - \sqrt{\sum_{uv} [\mathcal{A}_2^\phi + \mathbf{E}_2]_{uv}} \right| \\
&\le \frac{1}{n} \left( \sqrt{\left| \sum_{uv} [\mathcal{A}_1^\phi + \mathbf{E}_1]_{uv} - \sum_{uv} [\mathcal{A}_1^\phi]_{uv} \right|} + \sqrt{\left| \sum_{uv} [\mathcal{A}_2^\phi]_{uv} - \sum_{uv} [\mathcal{A}_2^\phi + \mathbf{E}_2]_{uv} \right|} \right) \\
&\le \frac{1}{n} \left( \sqrt{\left| \sum_{uv} [\mathbf{E}_1]_{uv} \right|} + \sqrt{\left| \sum_{uv} [\mathbf{E}_2]_{uv} \right|} \right) \\
&\le \frac{1}{n} \left( \sqrt{\|\mathbf{E}_1\|_1} + \sqrt{\|\mathbf{E}_2\|_1} \right) \\
&\le \frac{\sqrt{2}}{n} \sqrt{\|\mathbf{E}_1\|_1 + \|\mathbf{E}_2\|_1} \\
&\le \frac{\sqrt{2}}{n^{3/4}} \sqrt{\|\mathbf{E}_1\|_F + \|\mathbf{E}_2\|_F}
\end{aligned}
\tag{45}
$$

In the above derivation, the third inequality holds due to the fact that $|\sqrt{x} - \sqrt{y}| \le \sqrt{|x - y|}$ is true for any nonnegative $x$ and $y$.

Now we need to find bounds for $\|\mathbf{E}_1\|_F$ and $\|\mathbf{E}_2\|_F$.

$$
\begin{aligned}
\|\mathbf{E}_i\|_F &= \|\tilde{\mathbf{V}}_i \tilde{\mathbf{S}}_i \tilde{\mathbf{V}}_i^\top - \mathbf{V}_i \mathbf{S}_i \mathbf{V}_i^\top\|_F \\
&\le \|\tilde{\mathbf{V}}_i \tilde{\mathbf{S}}_i \tilde{\mathbf{V}}_i^\top - \mathbf{V}_i \tilde{\mathbf{S}}_i \tilde{\mathbf{V}}_i^\top\|_F + \|\mathbf{V}_i \tilde{\mathbf{S}}_i \tilde{\mathbf{V}}_i^\top - \mathbf{V}_i \mathbf{S}_i \tilde{\mathbf{V}}_i^\top\|_F + \|\mathbf{V}_i \mathbf{S}_i \tilde{\mathbf{V}}_i^\top - \mathbf{V}_i \mathbf{S}_i \mathbf{V}_i^\top\|_F \\
&\le \|\tilde{\mathbf{V}}_i - \mathbf{V}_i\|_F \|\tilde{\mathbf{S}}_i\|_2 \|\tilde{\mathbf{V}}_i\|_2 + \|\mathbf{V}_i\|_2 \|\tilde{\mathbf{S}}_i - \mathbf{S}_i\|_F \|\tilde{\mathbf{V}}_i\|_2 + \|\mathbf{V}_i\|_2 \|\mathbf{S}_i\|_2 \|\tilde{\mathbf{V}}_i - \mathbf{V}_i\|_F
\end{aligned}
\tag{46}
$$

It is known that

$$\|\mathbf{S}_i\|_2 = \sigma_1(\mathbf{A}_i), \quad \|\tilde{\mathbf{S}}_i\|_2 \le \sigma_1(\mathbf{A}_i) + \sigma_1(\boldsymbol{\Delta}_i), \quad \|\tilde{\mathbf{V}}_i\|_2 = \|\mathbf{V}_i\|_2 = 1 \tag{47}$$

According to the Weyl's inequality for singular values, we have

$$\|\tilde{\mathbf{S}}_i - \mathbf{S}_i\|_F \le \sqrt{n}\sigma_1(\boldsymbol{\Delta}_i) \tag{48}$$

According to the Corollary 3 of (Yu et al., 2015), a variant of the Davis-Kahan Theorem (Davis & Kahan, 1970), we have

$$\|\tilde{\mathbf{V}}_i - \mathbf{V}_i\|_F \le 2\sqrt{2n}\sigma_1(\boldsymbol{\Delta}_i)/\delta_i \tag{49}$$

where $\delta_i = \min_{j \ne k} |\sigma_j(\mathbf{A}_i) - \sigma_k(\mathbf{A}_i)|$. Combining (46), (48), and (49), we have

$$\begin{aligned}
\|\mathbf{E}_i\|_F &\le 2\sqrt{2n}\delta_i^{-1}\sigma_1(\boldsymbol{\Delta}_i)(2\sigma_1(\mathbf{A}_i) + \sigma_1(\boldsymbol{\Delta}_i)) + \sqrt{n}\sigma_1(\boldsymbol{\Delta}_i) \\
&\le 5\sqrt{2n}\delta_i^{-1}\sigma_1(\boldsymbol{\Delta}_i)\sigma_1(\mathbf{A}_i) + \sqrt{n}\sigma_1(\boldsymbol{\Delta}_i) \\
&= (5\sqrt{2}\delta_i^{-1}\sigma_1(\mathbf{A}_i) + 1)\sqrt{n}\sigma_1(\boldsymbol{\Delta}_i) \\
&\le 6\sqrt{2n}\delta_i^{-1}\sigma_1(\mathbf{A}_i)\sigma_1(\boldsymbol{\Delta}_i)
\end{aligned} \tag{50}$$

where the last inequality used the fact that $\sigma_1(\mathbf{A}_i) > \delta_i$. Combing (45) and (50), we arrive at

$$\begin{aligned}
&\left| \text{MMFD}(G_1', G_2') - \text{MMFD}(G_1, G_2) \right| \\
&\le \frac{\sqrt{2}}{n^{3/4}} \left( 6\sqrt{2n}\delta_1^{-1}\sigma_1(\mathbf{A}_1)\sigma_1(\boldsymbol{\Delta}_1) + 6\sqrt{2n}\delta_2^{-1}\sigma_1(\mathbf{A}_2)\sigma_1(\boldsymbol{\Delta}_2) \right)^{1/2} \\
&\le \frac{5}{n^{1/2}} \left( \delta_1^{-1}\sigma_1(\mathbf{A}_1)\sigma_1(\boldsymbol{\Delta}_1) + \delta_2^{-1}\sigma_1(\mathbf{A}_2)\sigma_1(\boldsymbol{\Delta}_2) \right)^{1/2}
\end{aligned} \tag{51}$$

$\square$

### D.5. Proof for Theorem 2.6

*Proof.* We have the following derivation for the difference between $\text{MMFD}_{\text{LR}}$ and MMFD.

$$\begin{aligned}
&\left| \text{MMFD}_{\text{LR}}(G_1, G_2) - \text{MMFD}(G_1, G_2) \right| \\
&= \left| \frac{1}{n}\sqrt{\sum_{uv}[\bar{\mathbf{A}}_1]_{uv} + \sum_{uv}[\bar{\mathbf{A}}_2]_{uv} - 2\sum_j \sigma_j(\bar{\mathbf{Q}})} - \frac{1}{n}\sqrt{\sum_{uv}[\mathbf{A}_1]_{uv} + \sum_{uv}[\mathbf{A}_2]_{uv} - 2\sum_j \sigma_j(\mathbf{Q})} \right| \\
&\stackrel{(a)}{\le} \frac{1}{n}\sqrt{\left| \sum_{uv}[\bar{\mathbf{A}}_1 - \mathbf{A}_1]_{uv} + \sum_{uv}[\bar{\mathbf{A}}_2 - \mathbf{A}_2]_{uv} + 2\left( \sum_j \sigma_j(\mathbf{Q}) - \sum_j \sigma_j(\bar{\mathbf{Q}}) \right) \right|} \\
&\le \frac{1}{n}\sqrt{\left| \sum_{uv}[\bar{\mathbf{A}}_1 - \mathbf{A}_1]_{uv} \right| + \left| \sum_{uv}[\bar{\mathbf{A}}_2 - \mathbf{A}_2]_{uv} \right| + 2\left| \sum_j \sigma_j(\mathbf{Q}) - \sum_j \sigma_j(\bar{\mathbf{Q}}) \right|} \\
&\le \frac{1}{n}\sqrt{\sum_{uv}\left[ |\bar{\mathbf{A}}_1 - \mathbf{A}_1| \right]_{uv} + \sum_{uv}\left[ |\bar{\mathbf{A}}_2 - \mathbf{A}_2| \right]_{uv} + 2\left| \sum_j \sigma_j(\mathbf{Q}) - \sum_j \sigma_j(\bar{\mathbf{Q}}) \right|}
\end{aligned} \tag{52}$$

In the above derivation, inequality (a) holds due to the fact that $|\sqrt{x} - \sqrt{y}| \le \sqrt{|x - y|}$ is true for any nonnegative $x$ and $y$.

In the theorem, the positive semi-definite assumption on $\mathbf{A}_i$ indicates that the EVD and SVD of $\mathbf{A}_i$ are exactly the same, which greatly simplifies the subsequent analysis, e.g., $\boldsymbol{\Phi}_i = \mathbf{S}_i^{1/2}\mathbf{V}_i^\top$ and $\boldsymbol{\Phi}_i = \bar{\mathbf{S}}_i^{1/2}\bar{\mathbf{V}}_i^\top$. We now derive the upper bound

for each term in the square root of the last inequality of (52). First, we have

$$
\sum_{uv} \left[ |\bar{\mathbf{A}}_1 - \mathbf{A}_1| \right]_{uv} = \| \bar{\mathbf{A}}_1 - \mathbf{A}_1 \|_1
$$

$$
\leq n \| \bar{\mathbf{A}}_1 - \mathbf{A}_1 \|_F \overset{(a)}{=} n \sum_{j=d+1}^{n} \sigma_j(\mathbf{A}_1)
\tag{53}
$$

where the equality (a) holds because $\bar{\mathbf{A}}$ is the best rank-$d$ approximation of $\mathbf{A}$. Similarly, we have

$$
\sum_{uv} \left[ |\bar{\mathbf{A}}_2 - \mathbf{A}_2| \right]_{uv} \leq n \sum_{j=d+1}^{n} \sigma_j(\mathbf{A}_2)
\tag{54}
$$

Regarding $\left| \sum_j \sigma_j(\mathbf{Q}) - \sum_j \sigma_j(\bar{\mathbf{Q}}) \right|$, we let $\tilde{\mathbf{S}}_i = \begin{bmatrix} \bar{\mathbf{S}}_i & \mathbf{0} \\ \mathbf{0} & \mathbf{0} \end{bmatrix} \in \mathbb{R}^{d \times d}$, $j = 1, 2$, and have

$$
\left| \sum_{j=1}^{n} \sigma_j(\mathbf{Q}) - \sum_{j=1}^{d} \sigma_j(\bar{\mathbf{Q}}) \right|
$$

$$
= \left| \sum_{j=1}^{n} \sigma_j(\mathbf{\Phi}_1 \mathbf{H} \mathbf{\Phi}_2^\top) - \sum_{j=1}^{d} \sigma_j(\bar{\mathbf{\Phi}}_1 \mathbf{H} \bar{\mathbf{\Phi}}_2^\top) \right|
$$

$$
= \left| \sum_{j=1}^{n} \sigma_j(\mathbf{S}_1^{1/2} \mathbf{V}_1^\top \mathbf{H} \mathbf{V}_2 \mathbf{S}_2^{1/2}) - \sum_{j=1}^{d} \sigma_j(\bar{\mathbf{S}}_1^{1/2} \bar{\mathbf{V}}_1^\top \mathbf{H} \bar{\mathbf{V}}_2 \bar{\mathbf{S}}_2^{1/2}) \right|
$$

$$
\overset{(a)}{=} \left| \sum_{j=1}^{n} \sigma_j(\mathbf{S}_1^{1/2} \mathbf{V}_1^\top \mathbf{H} \mathbf{V}_2 \mathbf{S}_2^{1/2}) - \sum_{j=1}^{n} \sigma_j(\tilde{\mathbf{S}}_1^{1/2} \mathbf{V}_1^\top \mathbf{H} \mathbf{V}_2 \tilde{\mathbf{S}}_2^{1/2}) \right|
$$

$$
\leq \sum_{j=1}^{n} \left| \sigma_j(\mathbf{S}_1^{1/2} \mathbf{V}_1^\top \mathbf{H} \mathbf{V}_2 \mathbf{S}_2^{1/2}) - \sigma_j(\tilde{\mathbf{S}}_1^{1/2} \mathbf{V}_1^\top \mathbf{H} \mathbf{V}_2 \tilde{\mathbf{S}}_2^{1/2}) \right|
$$

$$
\overset{(b)}{\leq} \sum_{j=1}^{n} \sigma_j \left( \mathbf{S}_1^{1/2} \mathbf{V}_1^\top \mathbf{H} \mathbf{V}_2 \mathbf{S}_2^{1/2} - \tilde{\mathbf{S}}_1^{1/2} \mathbf{V}_1^\top \mathbf{H} \mathbf{V}_2 \tilde{\mathbf{S}}_2^{1/2} \right)
$$

$$
= \sum_{j=1}^{n} \sigma_j \left( (\mathbf{S}_1^{1/2} - \tilde{\mathbf{S}}_1^{1/2}) \mathbf{V}_1^\top \mathbf{H} \mathbf{V}_2 \mathbf{S}_2^{1/2} + \tilde{\mathbf{S}}_1^{1/2} \mathbf{V}_1^\top \mathbf{H} \mathbf{V}_2 (\mathbf{S}_2^{1/2} - \tilde{\mathbf{S}}_2^{1/2}) \right)
$$

$$
\overset{(c)}{\leq} \sum_{j=1}^{n} \sigma_j \left( (\mathbf{S}_1^{1/2} - \tilde{\mathbf{S}}_1^{1/2}) \mathbf{V}_1^\top \mathbf{H} \mathbf{V}_2 \mathbf{S}_2^{1/2} \right) + \sum_{j=1}^{n} \sigma_j \left( \tilde{\mathbf{S}}_1^{1/2} \mathbf{V}_1^\top \mathbf{H} \mathbf{V}_2 (\mathbf{S}_2^{1/2} - \tilde{\mathbf{S}}_2^{1/2}) \right)
$$

$$
= \left\| (\mathbf{S}_1^{1/2} - \tilde{\mathbf{S}}_1^{1/2}) \mathbf{V}_1^\top \mathbf{H} \mathbf{V}_2 \mathbf{S}_2^{1/2} \right\|_* + \left\| \tilde{\mathbf{S}}_1^{1/2} \mathbf{V}_1^\top \mathbf{H} \mathbf{V}_2 (\mathbf{S}_2^{1/2} - \tilde{\mathbf{S}}_2^{1/2}) \right\|_*
$$

$$
\leq \left\| \mathbf{S}_1^{1/2} - \tilde{\mathbf{S}}_1^{1/2} \right\|_2 \| \mathbf{V}_1 \|_2 \| \mathbf{V}_2 \|_2 \left\| \mathbf{S}_2^{1/2} \right\|_2 \| \mathbf{H} \|_* + \left\| \mathbf{S}_2^{1/2} - \tilde{\mathbf{S}}_2^{1/2} \right\|_2 \| \mathbf{V}_1 \|_2 \| \mathbf{V}_2 \|_2 \left\| \mathbf{S}_1^{1/2} \right\|_2 \| \mathbf{H} \|_*
$$

$$
\overset{(d)}{=} n \sigma_{d+1}^{1/2}(\mathbf{A}_1) \sigma_1^{1/2}(\mathbf{A}_2) + n \sigma_{d+1}^{1/2}(\mathbf{A}_2) \sigma_1^{1/2}(\mathbf{A}_1)
\tag{55}
$$

In the derivation above, equality (a) holds because padding a matrix with zeros does not change the original singular values, inequality (b) holds according to the Theorem 3.4.5 in (Horn & Johnson, 1991), inequality (c) holds according to the Corollary 3.4.3 in (Horn & Johnson, 1991), and inequality (d) holds due to the fact that $\| \mathbf{V}_1 \|_2 = \| \mathbf{V}_2 \|_2 = 1$ and $\| \mathbf{H} \|_* = n$.

Combining (52), (53), (54), and (55), we arrive at

$$
\begin{aligned}
&\left| \mathrm{MMFD}_{\mathrm{LR}}(G_1, G_2) - \mathrm{MMFD}(G_1, G_2) \right| \\
&\leq \frac{1}{n} \sqrt{ n \sum_{j=d+1}^{n} \sigma_j(\mathbf{A}_1) + n \sum_{j=d+1}^{n} \sigma_j(\mathbf{A}_2) + 2\sigma_{d+1}^{1/2}(\mathbf{A}_1)\sigma_1^{1/2}(\mathbf{A}_2) + 2\sigma_{d+1}^{1/2}(\mathbf{A}_2)\sigma_1^{1/2}(\mathbf{A}_1) } \\
&= \frac{1}{n} \sqrt{ n \sum_{j=d+1}^{n} \left( \sigma_j(\mathbf{A}_1) + \sigma_j(\mathbf{A}_2) \right) + 2n\left( \sigma_1^{1/2}(\mathbf{A}_1)\sigma_{d+1}^{1/2}(\mathbf{A}_2) + \sigma_1^{1/2}(\mathbf{A}_2)\sigma_{d+1}^{1/2}(\mathbf{A}_1) \right) }
\end{aligned}
\tag{56}
$$

This completed the proof.

$\square$

### D.6. Proof for Theorem 2.8

*Proof.* It is obvious that $\mathrm{MFD}(G_1, G_2) \geq 0$, $\mathrm{MFD}(G_1, G_2) = \mathrm{MFD}(G_2, G_1)$, and $\mathrm{MFD}(G_1, G_1) = 0$ always hold. Here we only need to prove the triangle inequality. Let $\mathbf{R}_{ij}^* = \mathrm{argmin}_{\mathbf{R}_{ij} \in \mathcal{R}} \mathrm{MMD}\left( \boldsymbol{\Phi}_i, \mathbf{R}_{ij} \boldsymbol{\Phi}_j \right)$ and consider an arbitrary matrix $\mathbf{W} \in \mathbb{R}^{M \times M}$. Then based on the definition of MFD, we have

$$
\begin{aligned}
&\mathrm{MFD}(G_1, G_2) \\
&= \mathrm{MMD}\left( \boldsymbol{\Phi}_1, \mathbf{R}_{12}^* \boldsymbol{\Phi}_2 \right) \\
&\leq \mathrm{MMD}\left( \boldsymbol{\Phi}_1, \mathbf{W} \boldsymbol{\Phi}_2 \right) \\
&\leq \mathrm{MMD}\left( \boldsymbol{\Phi}_1, \mathbf{R}_{13}^* \boldsymbol{\Phi}_3 \right) + \mathrm{MMD}\left( \mathbf{R}_{13}^* \boldsymbol{\Phi}_3, \mathbf{W} \boldsymbol{\Phi}_2 \right)
\end{aligned}
\tag{57}
$$

When the kernel function used in MMD is a rotation-invariant kernel (e.g., polynomial kernel, radial basis function kernel, sigmoid kernel, or their combination), the computation of $\mathrm{MMD}\left( \mathbf{R}_{13}^* \boldsymbol{\Phi}_3, \mathbf{W} \boldsymbol{\Phi}_2 \right)$ is primarily based on the inner product $\boldsymbol{\Phi}_3^\top \mathbf{R}_{13}^{*\top} \mathbf{W} \boldsymbol{\Phi}_2$. Let $\mathbf{W}^\top \mathbf{R}_{13}^* = \mathbf{R}_{23}^*$, which has a solution $\hat{\mathbf{W}} = (\mathbf{R}_{23}^*(\mathbf{R}_{13}^*)^{-1})^\top = \mathbf{R}_{13}^*(\mathbf{R}_{23}^*)^\top$. We also have $\hat{\mathbf{W}}^\top \hat{\mathbf{W}} = \mathbf{R}_{23}^*(\mathbf{R}_{13}^*)^\top \mathbf{R}_{13}^*(\mathbf{R}_{23}^*)^\top = \mathbf{I}_M$, which means $\hat{\mathbf{W}} \in \mathcal{R}$. It follows that

$$
\boldsymbol{\Phi}_3^\top \mathbf{R}_{13}^{*\top} \hat{\mathbf{W}} \boldsymbol{\Phi}_2 = \boldsymbol{\Phi}_3^\top \mathbf{R}_{23}^{*\top} \boldsymbol{\Phi}_2
\tag{58}
$$

This means

$$
\mathrm{MMD}((\mathbf{R}_{13}^* \boldsymbol{\Phi}_3, \hat{\mathbf{W}} \boldsymbol{\Phi}_2) = \mathrm{MMD}\left( \mathbf{R}_{23}^* \boldsymbol{\Phi}_3, \boldsymbol{\Phi}_2 \right)
\tag{59}
$$

Combining (59) and (57) with $\mathbf{W}$ replaced by $\hat{\mathbf{W}}$, we obtain

$$
\mathrm{MFD}(G_1, G_2) \leq \mathrm{MMD}\left( \boldsymbol{\Phi}_1, \mathbf{R}_{13}^* \boldsymbol{\Phi}_3 \right) + \mathrm{MMD}\left( \mathbf{R}_{23}^* \boldsymbol{\Phi}_3, \boldsymbol{\Phi}_2 \right) = \mathrm{MFD}(G_1, G_3) + \mathrm{MFD}(G_2, G_3)
\tag{60}
$$

Currently, we cannot prove that $\mathrm{MFD}(G_1, G_2) \iff G_1 = G_2$. Therefore, MFD is a pseudo-metric. $\square$

### D.7. Proof for Theorem 2.9

*Proof.* For convenience, we define

$$
\bar{\mathcal{L}}(\mathbf{R}_{12}) := \mathcal{L}(\mathbf{R}_{12}^{(t-1)}) + \langle \mathbf{G}^{(t-1)}, \mathbf{R}_{12} - \mathbf{R}_{12}^{(t-1)} \rangle
\tag{61}
$$

where $\mathbf{G}^{(t-1)} = \nabla\mathcal{L}(\mathbf{R}_{12}^{(t-1)}) = 2\beta\sum_{uv}\alpha_{uv}\exp\left(2\beta\langle\phi_u^{(1)},\mathbf{R}_{12}^{(t-1)}\phi_v^{(2)}\rangle\right)\phi_u^{(1)}\phi_v^{(2)\top}$. We have the following derivation

$$
\begin{aligned}
&\mathcal{L}(\mathbf{R}_{12}) - \bar{\mathcal{L}}(\mathbf{R}_{12})\\
&= \sum_{u,v}\alpha_{uv}\exp\left(2\beta\langle\phi_u^{(1)},\mathbf{R}_{12}\phi_v^{(2)}\rangle\right) - \sum_{u,v}\alpha_{uv}\exp\left(2\beta\langle\phi_u^{(1)},\mathbf{R}_{12}^{(t-1)}\phi_v^{(2)}\rangle\right)\\
&\quad - \left\langle 2\beta\sum_{uv}\alpha_{uv}\exp\left(2\beta\langle\phi_u^{(1)},\mathbf{R}_{12}^{(t-1)}\phi_v^{(2)}\rangle\right)\phi_u^{(1)}\phi_v^{(2)\top}, \mathbf{R}_{12}-\mathbf{R}_{12}^{(t-1)}\right\rangle\\
&= \sum_{u,v}\alpha_{uv}\left(\exp\left(2\beta\langle\phi_u^{(1)},\mathbf{R}_{12}\phi_v^{(2)}\rangle\right) - \exp\left(2\beta\langle\phi_u^{(1)},\mathbf{R}_{12}^{(t-1)}\phi_v^{(2)}\rangle\right)\right.\\
&\quad \left. - \left\langle 2\beta\exp\left(2\beta\langle\phi_u^{(1)},\mathbf{R}_{12}^{(t-1)}\phi_v^{(2)}\rangle\right)\phi_u^{(1)}\phi_v^{(2)\top}, \mathbf{R}_{12}-\mathbf{R}_{12}^{(t-1)}\right\rangle\right)\\
&= \sum_{u,v}\alpha_{uv}\exp\left(2\beta\langle\phi_u^{(1)},\mathbf{R}_{12}^{(t-1)}\phi_v^{(2)}\rangle\right)\left(\exp\left(2\beta\langle\phi_u^{(1)},\mathbf{R}_{12}-\mathbf{R}_{12}^{(t-1)}\phi_v^{(2)}\rangle\right)\right.\\
&\quad \left. - \left(1+2\beta\langle\phi_u^{(1)},\mathbf{R}_{12}-\mathbf{R}_{12}^{(t-1)}\phi_v^{(2)}\rangle\right)\right)\\
&\geq 0
\end{aligned}
\tag{62}
$$

where the inequality holds because $\alpha_{uv}\exp\left(2\beta\langle\phi_u^{(1)},\mathbf{R}_{12}^{(t-1)}\phi_v^{(2)}\rangle\right) > 0$ and $\exp(x) > 1+x$ for any $x$.

Now we define

$$
\hat{\mathcal{L}}(\mathbf{R}_{12}) = \bar{\mathcal{L}}(\mathbf{R}_{12}) - \frac{\rho}{2}\|\mathbf{R}_{12}-\mathbf{R}_{12}^{(t-1)}\|_F^2
\tag{63}
$$

which is exactly the right-hand-side of (23). Note that $\mathcal{L}(\mathbf{R}_{12}^{(t-1)}) = \bar{\mathcal{L}}(\mathbf{R}_{12}^{(t-1)}) = \hat{\mathcal{L}}(\mathbf{R}_{12}^{(t-1)})$.

Combining (62) and (63) yields

$$
\mathcal{L}(\mathbf{R}_{12}) \geq \bar{\mathcal{L}}(\mathbf{R}_{12}) = \hat{\mathcal{L}}(\mathbf{R}_{12}) + \frac{\rho}{2}\|\mathbf{R}_{12}-\mathbf{R}_{12}^{(t-1)}\|_F^2
\tag{64}
$$

As the inequality holds for any $\mathbf{R}_{12} \in \mathcal{R}$, letting $\mathbf{R}_{12} = \mathbf{R}_{12}^{(t)}$, we have

$$
\mathcal{L}(\mathbf{R}_{12}^{(t)}) \geq \hat{\mathcal{L}}(\mathbf{R}_{12}^{(t)}) + \frac{\rho}{2}\|\mathbf{R}_{12}^{(t)}-\mathbf{R}_{12}^{(t-1)}\|_F^2
\tag{65}
$$

Since $\mathbf{R}_{12}^{(t)}$ maximize $\hat{\mathcal{L}}(\mathbf{R}_{12})$, it holds that

$$
\hat{\mathcal{L}}(\mathbf{R}_{12}^{(t)}) \geq \hat{\mathcal{L}}(\mathbf{R}_{12}^{(t-1)})
\tag{66}
$$

Combining (66), (65), and the fact $\hat{\mathcal{L}}(\mathbf{R}_{12}^{(t-1)}) = \mathcal{L}(\mathbf{R}_{12}^{(t-1)})$, we arrive at

$$
\mathcal{L}(\mathbf{R}_{12}^{(t)}) \geq \mathcal{L}(\mathbf{R}_{12}^{(t-1)}) + \frac{\rho}{2}\|\mathbf{R}_{12}^{(t)}-\mathbf{R}_{12}^{(t-1)}\|_F^2
\tag{67}
$$

This proved part (a) of the theorem.

For part (b) of the theorem, we sum up both sides of (67) from 1 to $t$ and obtain

$$
\mathcal{L}(\mathbf{R}_{12}^{(t)}) - \mathcal{L}(\mathbf{R}_{12}^{(0)}) \geq \frac{\rho}{2}\sum_{i=1}^{t}\|\mathbf{R}_{12}^{(i)}-\mathbf{R}_{12}^{(i-1)}\|_F^2
\tag{68}
$$

Note that $\mathcal{L}(\mathbf{R}_{12}^{(\infty)}) < \infty$ and $\mathcal{L}(\mathbf{R}_{12}^{(0)}) \geq 0$, according to the definition of $\mathcal{L}(\mathbf{R}_{12})$. Therefore,

$$
\frac{\rho}{2}\sum_{i=0}^{\infty}\|\mathbf{R}_{12}^{(i)}-\mathbf{R}_{12}^{(i-1)}\|_F^2 < \infty
\tag{69}
$$

Since $\rho > 0$, we conclude from (69) that

$$
\mathbf{R}_{12}^{(i)}-\mathbf{R}_{12}^{(i-1)} \to \mathbf{0} \quad\text{when } i \to \infty
\tag{70}
$$

This completed the proof for part (b) of the theorem. $\square$

