# OpenReview forum: "Graph Minimum Factorization Distance and Its Application to Large-Scale Graph Data Clustering"
_ICML.cc/2025/Conference — ICML 2025 poster_

### Official Review · Reviewer_wUZK · 2025-03-07

**Overall Recommendation:** 4

**Summary:**

The work proposed a distance metric MMFD between graphs through comparing distributions and showed that MMFD is a pseudo metric and has a closed-form solution. The work also proposed several variants of MMFD and analyzed the properties theoretically. The proposed methods were compared with graph kernels, graph neural networks, and other distances in synthetic data analysis and graph clustering and showed improved performance.

## update after rebuttal
I appreciate the details and clarifications provided by the authors. I have no more concerns and will keep the rating.

**Claims And Evidence:**

Yes.

**Essential References Not Discussed:**

N/A

**Ethics Expertise Needed:**

["Other expertise"]

**Experimental Designs Or Analyses:**

Yes. The design and analysis of graph comparing and clustering are reasonable.

**Methods And Evaluation Criteria:**

Yes.

**Other Comments Or Suggestions:**

The paper will be stronger if my questions could be addressed successfully. In addition, the author may consider shortening the introduction section or other sections and move some results from the appendix to the main paper.

**Other Strengths And Weaknesses:**

Strengths:
1.The algorithms proposed in the paper are novel and elegant.
2.The presentations like the figures and tables are nice.
3.The proposed methods are compared with many baselines and the improvements are remarkable.
Weaknesses:
The explanations for several claims or results are not sufficient. Please refer to my questions.

**Questions For Authors:**

1.In the text below Theorem 2.3, it was said that “...where MMFDs between graphs in the same cluster should be small...” How does this relate to the perturbations?
2.On page 4, it is not clear why the MMFD between two complete graphs is always zero. Could the authors provide more explanation or theoretical justification?
3.Section 2.5 compared MMFD with MMD. Is MMD always greater than MMFD? In addition to the listed two issues of SVD, is it possible that the order of singular values has some error caused by noise? Could the authors provide show some examples of matrix R12? Readers would like to see how R12 differs from an identity matrix or sign-flip matrix, although the final computation of MMFD is not explicitly related to R12.
4.As the MFD proposed in Section 2.8 has no closed-form solution, it is necessary to analysis its time complexity.
5.In Table 4, it is surprising that the time costs of the WL kernel and MMFD are less than one second, as the complexity is $O(N2)$ if I understand correctly. Could the authors provide more explanation about these tiny time costs?

**Relation To Broader Scientific Literature:**

The proposed distances and clustering algorithms are useful.

**Theoretical Claims:**

Yes, I briefly checked the proof of the theorems.

---

> ### Author Rebuttal · Authors · 2025-03-28
>
> Dear Reviewer,
>
> We sincerely appreciate your recognition of our work. Our responses to your comments are as follows.
>
> **To Q1:**
>
> In a dataset, the graphs belonging to the same cluster are usually similar to each other. Therefore, for a pair of very similar graphs, denoted as $G_1$ and $G_2$, we can regard $G_2$ as a perturbed graph from $G_1$， and vice versa. Therefore, $\text{MMFD}(G_1,G_2)$ will be small. It also means that within-cluster distances are small, which is important for downstream tasks.
>
> **To Q2:**
>
> Your this question is the same as the question 4 asked by Reviewer 8oX9. We provide the following theoretical justification.
>
> A complete graph is a graph in which each vertex is connected to every other vertex. Therefore, for two complete graphs, their self-looped adjacency matrices are $\mathbf{A} _i=[1] _{n _i\times n _i}$, $i=1,2$.
> $\mathbf{A} _i$ are PSD and rank-1. We have $\boldsymbol{\Phi} _i=s\cdot[1] _{1\times n _i}$, where $s$ is $-1$ or $1$. That means $\boldsymbol{\mu} _i=-1$ or $+1$. The rotation matrix $\mathbf{R} _{12}$ is now just a scalar equal to $-1$ or $+1$. Therefore, $\min _{\mathbf{R} _{12}\in\mathcal{R}} \Vert \boldsymbol{\mu} _1-\mathbf{R} _{12}\boldsymbol{\mu} _2 \Vert\equiv 0$, for any $(n _1,n _2)$.
>
> **To Q3:**
>
> This is an insightful question. Yes, MMD is always larger than or equal to MMFD.
> In addition to the listed two issues of SVD, it is indeed possible that the order of singular values has some error caused by noise. We will include this in the revised paper. Regarding $\mathbf{R} _{12}$, we take the graphs $G_1$ and $G_7$ in Table 2 of our paper as an example. Let $d=4$, we have the following $\mathbf{R} _{12}$. We can see that it is quite different from an identity matrix or sign-flip matrix.
> \begin{equation}
>  \begin{matrix}
>  \hline
>      0.9985   & 0.0000  & -0.0087 &  -0.0538 \\\\
>    -0.0001    &1.0000   &-0.0005  & -0.0001 \\\\
>     0.0087   & 0.0005   & 1.0000   & 0.0001 \\\\
>     0.0538   & 0.0001   &-0.0006   & 0.9985  \\\\ \hline
> \end{matrix}
> \end{equation}
>
> **To Q4:**
>
> The time complexity of MFD is $\mathcal{O}\left(n^2 \log (d)+d^2 n+T\left(d^3+d n^2\right)\right)$, where $n$ is the number of nodes, $d$ is the rank, and $T$ is the number of iterations in the optimization. This complexity is much higher than that of MMFD and is mainly due to $Tdn^2$. Actually, we have a faster MFD based clustering algorithm, called MFD-KD. The time complexity of MFD-KD is $\mathcal{O}\left(n^2 \log (d)+d^2 n+T\left(d^3+d n_s^2\right)\right)$, where $n_s$ is much less $n$.
>
> **To Q5:**
>
> Thanks for the question. The tiny time costs of WL kernel and MMFD are due to the following reasons:
>  * In Table 4, we recorded only the time costs of computing the distance or similarity matrices on the two datasets, which was declared in line 432, column 2. That means we didn't include the time cost of spectral clustering.
>  * The experiments were conducted on a computer with Intel Core i9-12900K and RAM 64GB, which has a quite good CPU.
>  * The two datasets are not too large.
>  * WL kernel and MMFD are both efficient.
>
> These four reasons make the time costs of WL kernel and MMFD less than one second. We assure that the result is correct and reproducible.
>
> **Thank you again for your comments and suggestions. We are looking forward to your feedback.**

---

> > ### Comment · Reviewer_wUZK · 2025-04-04
> >
> > I appreciate the details and clarifications provided by the authors. I have no more concerns  and will keep the rating.

---

> > > ### Author Response · Authors · 2025-04-04
> > >
> > > We really appreciate your feedback and support. We have added the details to the paper.

---

### Official Review · Reviewer_rKce · 2025-03-10

**Overall Recommendation:** 5

**Summary:**

The paper studies graph distance and graph clustering. It introduced a minimum mean factor distance (MMFD) between graphs and its extensions such as low-rank MMFD, MMFD-KM, and MFD.Theses methods outperformed many baselines on large-scale graph datasets such as REDDIT-5K.

**Claims And Evidence:**

The claims are supported by clear and convincing evidence.

**Essential References Not Discussed:**

The literature review seems complete.

**Experimental Designs Or Analyses:**

Yes, I checked the experimental settings, evaluation metrics, and analysis.

**Methods And Evaluation Criteria:**

Yes, the proposed methods and evaluation make sense.

**Other Comments Or Suggestions:**

See my questions.

**Other Strengths And Weaknesses:**

Strengths:

* The paper proposed an effective distance metric MMFD for comparing graphs. MMFD has a closed-form solution and hence is much more efficient than other methods such as graph kernels, GWD, and GED.

* Some extensions such MMFD-KM and MFD were developed.An approach to incorporating node attributes was provided.

* The paper has some theoretical results to support the proposed methods.

* In the experiments, the proposed methods outperformed other methods with a large margin.

Weaknesses:

* The time cost comparison is only conducted on smaller datasets.In Table 4, the two datasets are not large.

* The setting or analysis for the hyperparameters of the proposed methods seems missing.

**Questions For Authors:**

* In Table 1, what do K and T mean?

* Are the proposed methods applicable to directed graphs? I am just curious about this because Theorem 2.2 mentioned “undirected graphs”.

* In (20), how to set $\lambda$ for graph classification and spectral clustering?

* It is better to show some time cost comparison on larger datasets.

* In the caption of Table 3, the explanation for the colored text seems wrong. The purple color corresponds to best result in each case?

* How do the hyperparameters of the proposed methods affect their performance?

* It is suggested to provide more discussion about the experimental results in Table 3. For instance, on the AIDS dataset, the NMI and ARI of the proposed methods are 10 or 20 points higher than other methods. What is the possible reason for this phenomenon?

**Relation To Broader Scientific Literature:**

Comparing graphs is a fundamental problem in machine learning.

**Theoretical Claims:**

Yes, I checked the proofs for theoretical claims such as Theorems 2.2 and 2.3.

---

> ### Author Rebuttal · Authors · 2025-03-28
>
> Dear Reviewer,
>
> We are grateful for your acknowledgment of our work. Our responses to your comments are as follows.
>
> **To W1:**
>
> Thanks for pointing it out. Actually, we have the time cost comparison on much larger datasets (e.g. graphs with 10000 nodes). They are in Table 8 of Appendix C4. By the way, some baselines such as Entropic GWD (shown by Table 2) are too time-consuming and hence cannot be applied to very large datasets such as the two REDDIT datasets.
>
> **To W2:**
>
> The settings of the hyperparameters are in Appendix B.2. Some results of hyperparameters are in Tables 6 and 7. For instance, our MMFD is not sensitive to the factorization dimension $d$. We will provide more analysis on them.
>
> **To Q1:**
>
> In Table 1, K is the number of clusters and T is the number of iterations in the clustering algorithm MMFD-KM. We are sorry about the absence of the description.
>
> **To Q2:**
>
> We think our methods can be applied to directed graphs, though we haven't conducted any experiments related to directed graphs. We may consider the following strategy. Given a directed graph, with adjacency matrix $\mathbf{A}$, which is asymmetric, we perform SVD $\mathbf{A}=\mathbf{USV}^\top$ and generate the following augmented adjacency matrix $\bar{\mathbf{A}}=[\mathbf{USU}^\top, \boldsymbol{0};\boldsymbol{0},\mathbf{VSV}^\top]$. $\bar{\mathbf{A}}$ is PSD and hence can be used in our methods such as MMFD and MFD.
>
> **To Q3:**
> Do you mean the $\gamma$ in (20)? In all experiments, we set it to $1 /(5 u)^2$, where $u$ denotes the average MMDF (or MMD, GWD, and GED) between all graphs. This setting was explained in line 837, Appendix B2.
>
> **To Q4:**
>
> The following table presents some results of time cost (second) comparison on very large graphs. We can see that the Shortest-path kernel is most time-consuming, while our MMFD is always the most efficient one.
> \begin{equation}
>     \begin{matrix}
> \hline n \text{(number of nodes)} & 1000 & 2000 & 4000 & 6000  & 8000 & 10000 & 20000 & 30000 \\\\
> \hline \text { Shortest-path kernel }  & 4.853 & 25.8  & 178.7  & 524.8  & 1125.0 & 2022.4 & - & - \\\\
> \text {WL subtree kernel}  & 0.393 & 1.5 &  6.6 & 15.3  & 27.4 & 43.4 & 182.6 & 449.7 \\\\
> \text {entropic Gromove Wasserstein}  & 0.367 & 0.5 &  2.3 &  6.4  & 12.8 & 21.9 & 169.1 & 579.7 \\\\
> \text {MMFD}_{\text{LR}} & \mathbf{0 . 0 9 5} & \mathbf{0 . 3} & \mathbf{1 . 1} &  \mathbf{2 . 1} & \mathbf{3 . 7} & \mathbf{6 . 4} & \mathbf{2 9 . 9} & \mathbf{7 8 . 8} \\\\
> \hline
> \end{matrix}
> \end{equation}
>
>
> **To Q5:**
>
> Thanks for pointing out the mistakes. We have corrected them.
>
> **To Q6:**
>
> Our MMFD has only one hyperparameter $d$. As shown in Table 7, we compared the performance of MMFD with different $d$ on datasets PROTEINS and REDDIT-MULTI-5K. On PROTEINS, a larger $d$ is better since most graphs are very small. On REDDIT-MULTI-5K, A smaller $d$ can improve the clustering performance slightly, since most graphs are very large. In sum, MMFD is not sensitive to $d$.
>
> Regarding the extension method MFD, besides $d$, there is a hyperparameter $\rho$ in the optimization algorithm and a hyperparameter $\beta$ in the kernel function.  Figure 3 shows that the optimization is not sensitive to $\rho$. For $\beta$, we just set it as the inverse of the squared average distance in the experiments, which works well.
>
> **To Q7:**
>
> In the revised paper, we have added more discussion about the results. Regarding the improvement on the dataset AIDS, one possible reason is that the distribution information in the dataset is more useful than other datasets, and our methods MMFD and MFD can effectively exploit the information.
>
> **We thank you again for your comments and time. We are looking forward to your response.**

---

> > ### Comment · Reviewer_rKce · 2025-04-03
> >
> > I appreciate the answers and clarification. I have no concerns about the work and hence keep the rating.

---

> > > ### Author Response · Authors · 2025-04-03
> > >
> > > We are extremely grateful for your response to our rebuttal and your acknowledgment of our work.

---

### Official Review · Reviewer_v4JQ · 2025-03-12

**Overall Recommendation:** 3

**Summary:**

This paper introduces a new measurement, named MMFD, for comparing and clustering graph data. By considering the adjacency matrix of graph as a kernel matrix, MMFD transforms the graph comparison problem into distribution comparison.
This paper then proposes a low-rank approximation and a generalized version algorithm for large graph efficient clustering.

**Claims And Evidence:**

Yes

**Essential References Not Discussed:**

No

**Experimental Designs Or Analyses:**

Yes

**Methods And Evaluation Criteria:**

Yes

**Other Comments Or Suggestions:**

No

**Other Strengths And Weaknesses:**

Strengths
- highly scalable and suitable for large datasets.
- superior clustering performance, as evidenced by the experiments on several real-world datasets, outperforming competitors in terms of clustering accuracy and time.

**Questions For Authors:**

- As mentioned by the authors, although the low-order approximation leads to some loss of information, it has a denoising effect. I'm interested if there are relevant toy experiments proving this yet.
- Can MMFD cope with missing attributes?
- MMFD is highly efficient and theoretically guaranteed. I would like to know how much its performance gap is compared with UGRL? In addition, I would like to know about the real application tasks of graph comparison, such as chemical molecular property prediction?

**Relation To Broader Scientific Literature:**

The key contributions of the paper are related to the broader scientific literature through existing graph comparing methods. The paper builds on past research while offering improvements, particularly in efficiency and scalability.

**Theoretical Claims:**

I didn't check the proofs for the theoretical claims.

---

> ### Author Rebuttal · Authors · 2025-03-28
>
> Dear Reviewer,
>
> It is our great honor to receive your positive assessment. Our responses to your questions are as follows.
>
> **To Q1:**
>
> Yes. Take $G_1$ and $G_3$ in Table 2 of our paper as an example, where $\text{MMFD}(G_1,G_3)=0.1598$. We add one additional node to $G_1$ and the additional node is only connected with one node. The same operation is conducted on $G_2$. Then we get two noisy graphs $G_1'$ and $G_3'$, where the noises are very strong because the graphs are too small. We compute the low-rank MMFD (with different rank) between $G_1'$ and $G_3'$. The difference between $\text{MMFD} _{\text{LR}}(G _1',G _3')$ and $\text{MMFD}(G _1,G _3)$ are shown in the following table. We see that when the rank decreases from 6 to 3, the difference becomes smaller. This demonstrates the denoising effect.
> Additionally, as shown in Table 7 of our paper, on dataset REDDIT-MULTI-5K, using a lower rank (but not too low) yields a better clustering performance, which also demonstrates the potential denoising effect.
>
> \begin{equation}
> \begin{matrix}
> \hline
> \text{rank} & \text{full} & 5 &4 &3 &2 &1\\\\ \hline
>     |\text{MMFD}(G_1,G_3)-\text{MMFD}_{\text{LR}}(G_1',G_3')| & 0.073 & 0.070& 0.049& \textbf{0.044} & 0.046  & 0.047  \\\\     \hline
> \end{matrix}
> \end{equation}
>
> **To Q2:**
>
> Our MMFD is mainly designed for comparing the adjacency matrices of two graphs, though it is able to utilize node attributes, as explained in Section 2.6. To apply MMFD to graphs with incomplete node attribute matrices, we need to do missing data imputation (e.g. matrix completion) in advance. Therefore, MMFD cannot directly handle missing attributes. However, here we provide a possible approach. Let $\tilde{G}_1$ and $\tilde{G}_2$ be the two augmented graphs using node attributes, which is explained in Section 2.6. We just denote the missing values of node attribute matrices as $x_1$ and $x_2$, respectively, and write $\tilde{G} _1=\tilde{G} _1(x _1)$ and $\tilde{G} _2=\tilde{G} _2(x _2)$. Then, according to the definition of MMFD, we can solve $\min _{x _1,x _2}\text{MMFD}(\tilde{G} _1(x _1),\tilde{G} _2(x _2))$ to impute the missing values and get the minimum distance. Nevertheless, the optimization and effectiveness need further investigation, which could be a future work.
>
> **To Q3(a):**
>
> In Table 3 of our paper, we compared our MMFD with four pure UGRL methods including InfoGraph, GraphCL, JOAO, and GWF and two clutering-oriented UGRL methods including GLCC and DCGLC. On dataset AIDS, the best NMI of UGRL methods is $76\%$, while the NMI of our MMFD is $88\%$, meaning that the gap is significantly large. For convenience, we calculate the average of ACC, NMI, and ARI of each method and take the average over the 2 datasets (PROTEIN and AIDS), the results are compared in the following table. Our MMFD and MFD outperformed the best competitor by $6\%$.
>
> \begin{equation}
>  \begin{matrix}
>  \hline
> \text{Method} &\text{InfoGraph} & \text{GraphCL} & \text{JOAO} & \text{GWF} & \text{GLCC} & \text{DCGLC} & \text{MMFD} & \text{MFD} \\\\
> \text{Metric (\\%)} &42 &40 &17 &35 &20 &47 &\textbf{53} &\textbf{54} \\\\
> \hline
> \end{matrix}
> \end{equation}
>
> We also compare the time costs of DCGLC (SOTA of graph clustering) and our MMFD-KM and MFD-KD on dataset REDDIT-12K in the following table.
>
> \begin{equation}
>  \begin{matrix}
>  \hline
> \text{Method} &\text{DCGLC} & \text{MMFD-KM} & \text{MFD-KD} \\\\
> \text{Time cost} & 21\ hours & 2.2\ minutes & 3\ hours \\\\
> \hline
> \end{matrix}
> \end{equation}
>
> Besides the advantage of higher efficiency and accuracy, our MMFD has nearly no hyperparameter to tune (it is not sensitive to $d$), while deep learning methods have quite a few hyperparameters that are very difficult to tune in unsupervised learning.
>
> **To Q3(b):**
>
> Regarding chemical molecular property prediction, we combine MMFD with support vector regression and call this combination MMFD+SVR. We apply it to the QM9 dataset. Since kernel SVR is not scalable to very large datasets, we only use 25000 graphs for training and 5000 graphs for testing. Some results (space limitation) of MAE are in the following table, where the classic MPNN and enn-s2s [1] are compared. We see MMFD+SVR works well. It still has a lot of room for improvement, e.g., by using attributes more effectively, using the extension MFD, or using more training data.
>
> \begin{equation}
>  \begin{matrix}
>  \hline
> \text{Target} & \mu & \alpha & \text{HOMO} & \text{LUMO} & \text{gap} & \text{R2} & \text{ZPVE} &\text{U0}\\\\ \hline
> \text{MPNN}& 1.22 & 1.55 & 1.17 & 1.08 & 1.70 &3.99 &2.52 & 3.02\\\\
> \text{enn-s2s} & \textbf{0.30} & 0.92 & 0.99 & 0.87& 1.60& \textbf{0.15} & 1.27& 0.45\\\\
> \text{MMFD+SVR} &0.64 & \textbf{0.34}& \textbf{0.64}& \textbf{0.43}& \textbf{0.43} & 0.54 & \textbf{0.07}&\textbf{0.41}\\\\
> \hline
> \end{matrix}
> \end{equation}
> [1] Gilmer et al. Neural Message Passing for Quantum Chemistry.
>
> **Thank you again. We are looking forward to your feedback.**

---

> > ### Comment · Reviewer_v4JQ · 2025-04-04
> >
> > I appreciate the detailed information provided by the authors. I have no more concerns and will keep the rating.

---

> > > ### Author Response · Authors · 2025-04-04
> > >
> > > Thank you so much for your feedback on our rebuttal and acknowledgment of our method's effectiveness. We have added the additional results to the paper.
> > >
> > > **If you think there is no weakness in our work, could you please raise the rating appropriately?**

---

### Official Review · Reviewer_8oX9 · 2025-03-13

**Overall Recommendation:** 3

**Summary:**

The authors study the clustering problem for graph data. They propose a new measure between two graphs called minimum mean factor distance (MMFD) that serves as a kernel function in graph data clustering. MMFD measure the minimum distance (through rotation by a real orthonormal matrix) of the mean vectors of $\Phi_1$ and $\Phi_2$ that can be viewed as the feature vectors of nodes from two graphs $G_1$ and $G_2$, respectively. Two variants of MMFD, named Low-Rank MMFD and MF, are given to lower the time complexity and to fully exploit $\Phi_1$ and $\Phi_2$, respectively. The experiments evaluate the effectiveness and efficiency of their methods.

=========================

After reading the authors' rebuttals and rethinking the significance of MMFD, I agree that MMFD has a certain value, although I still cannot understand why if the original self-looped adjacency matrices are both PSD, the distance could be zero once the sums of elements in two matrices are equal. In this context, MFD seems more reasonable. I can only say that the instance in Table 2 and the experimental results look convincing. Moreover, I hope the authors can reorganize Section 2.3 due to my concern of W1. I decide to raise my score to 3 (Weak accept). Thank the authors for the responses.

**Claims And Evidence:**

All claims are supported by evidence, but some descriptions are not so clear. Please refer to the questions below.

**Essential References Not Discussed:**

I think the related works included in the paper have been essential to understanding the key contributions.

**Ethical Review Concerns:**

No ethical review concern.

**Experimental Designs Or Analyses:**

Yes, I have checked the soundness/validity of experimental designs and analyses.

**Methods And Evaluation Criteria:**

Yes, I think the methods and evaluation criteria make sense.

**Other Comments Or Suggestions:**

1. Please give the full names of EVD and SVD when they appear for the first time.
2. Line 206, Column 2, please clarify the inner product $\langle A, B \rangle=\text{trace}(A^T B)$.
3. “$i=1, 2$” should be deleted in Line 134, Column 2.

**Other Strengths And Weaknesses:**

S1. The MMFD measure performs well in evaluation. The toy example in Table 2 is intuitive and convincing.

W1. The authors have spent much space on showing that MMFD is only related to $\mathcal{A}_i^\phi$’s. But it can be observed easily from Eq. (10) or Eq. (11) that MMFD measures actually the difference of the lengths of $\mu_1$ and $\mu_2$, that is the absolute value of $\Vert\mu_1\Vert-\Vert\mu_2\Vert$ (when they are parallel by rotation). A plain expansion of $\left| \Vert\mu_1\Vert-\Vert\mu_2\Vert \right|$ yields Eq. (16).

W2. There are a lot of unclear descriptions in the main text. Please refer to the questions below. Because of this, I don’t think this paper is suitable for publication with present version.

**Questions For Authors:**

1. Line 212, Column 1, why do you need to make the distance as small as possible and call it a natural principle? This seems the core of MMFD, and I can feel it to some extent, but need more explanations.
2. What is the difference between Eqs. (3) and (7)? Do you mean that if we have the PSD matrices $A_i$’s, then it’s good, but in real applications we do not have, so you need to construct the PSD matrices $\mathcal{A}_i^\phi$’s that approximate of $A_i$’s? Are $A_i$ and its augmentation PSD in Alg. 1? Is $\mathcal{A}_i^\phi$ necessary when $A_i$ and its augmentation are PSD?
3. How do you perform the randomized SVD?
4. You have assumed $G_1$ and $G_2$ of the same size in Line 134, Column 1. But it seems that $n_1$ and $n_2$ could be different in Alg 1. Do you define MMFD between two graphs of different sizes as that in Line 10 of Alg. 1? Why can you define it like that? Why can you say assuming $n_1=n_2$ is without loss of generality? Why is MMFD between two complete graphs of different sizes (Line 195, Column 2) zero?
5. What is the $0.5$ in Line 296, Column 1? The probability in randomized SVD? What does it mean?

**Relation To Broader Scientific Literature:**

This paper fits in the study line of clustering of graphs. It proposes a new kernel function and some variants, which is a contribution to graph clustering and kernel methods.

**Theoretical Claims:**

I have checked the proofs in the main text. Some parts of the derivation process on the basic MMFD are lengthy and redundant, which makes not too much sense. Please refer to the weakness.

---

> ### Author Rebuttal · Authors · 2025-03-28
>
> Dear Reviewer,
>
> We sincerely appreciate your insightful comments. They improved our work a lot.
>
> **To W1:**
>
> Previously, we focused on deriving from Equation (12) and overlooked Equation (11). Your advice has helped us make Section 2.3 more concise, thereby freeing up more space to supplement other important results or discussions.
>
> **To W2:**
>
> We are trying our best to make the corresponding descriptions clearer and believe that the issues can be addressed by revision easily. Please refer to our responses to your questions.
>
> **To Other Comments Or Suggestions:**
>
> Thanks for your careful reading. We have revised our paper to resolve these issues.
>
> **To Q1:**
>
> As mentioned in Section 3.2, $\boldsymbol{\Phi}_1$ and $\boldsymbol{\Phi}_2$ cannot be uniquely determined, so their distances are not unique.
> We used the idea of ``shortest distance", which is quite general in real problems. Suppose there are three cities A, B, and C, and there exist multiple paths between each pair of them. To compare the distance between A and B and the distance between A and C, we use the shortest path between A and B and the shortest path between A and C, rather than any other longer paths.
>
> Back to our problem, for a graph $G_i$,  $\boldsymbol{\Phi}_i$ and $\bar{\boldsymbol{\Phi}}_i:=\mathbf{R}_i \boldsymbol{\Phi}_i$ are equivalent in terms of representing the adjacency matrix, $i=1,2$. The non-uniqueness of $\boldsymbol{\Phi}_i$ leads to a set of possible distances, denoted as $\mathcal{S}:=\\{\Vert \text{Mean}(\mathbf{R}_1\bar{\boldsymbol{\Phi}}_1)-\text{Mean}(\mathbf{R}_2\bar{\boldsymbol{\Phi}}_2)\Vert: \mathbf{R}_1, \mathbf{R}_2 \in \mathcal{R}\\}=\\{\Vert \mathbf{R}_1\boldsymbol{\mu}_1-\mathbf{R}_2\boldsymbol{\mu}_2\Vert: \mathbf{R}_1, \mathbf{R}_2 \in \mathcal{R}\\}$, where the second equality is due to that we can exchange $\boldsymbol{\Phi}_i$ and $\bar{\boldsymbol{\Phi}}_i$. We then define the smallest distance in $\mathcal{S}$ as the final distance between $G_1$ and $G_2$, i.e., $\text{MMFD}\left(G_1, G_2\right)=\min (S)$; it is equivalent to optimize $\mathbf{R}_1$ and $\mathbf{R}_2$ to minimize $\Vert \mathbf{R}_1\boldsymbol{\mu}_1-\mathbf{R}_2\boldsymbol{\mu}_2\Vert$.
>
> Please feel free to let us know if this explanation is not sufficient.
>
> **To Q2:**
>
> 1) Eqs. (3) is the ideal case and usually does not hold for real graphs because their adjacency matrices are often not PSD. Eqs. (7) is based on the PSD $\mathcal{A}_i^\phi$ and hence always holds for real graphs. $\mathcal{A}_i^\phi$ can be regarded as an PSD proxy or approximation of $\mathbf{A}_i$.
>
> 2) Yes. Your understanding is correct.
>
> 3) Yes. In Alg.1, $\mathbf{A}_i$ is the input, and the augmentation $\mathcal{A}_i^\phi$ is constructed by line 6 or line 8.
>
> 4) Does the "augmentation" you mentioned mean the one defined in equation (5)? In the algorithm, we use (5) rather than (4), which is stated in line 171 and line 4-8 of the algorithm. If $\mathbf{A}_i$ is PSD, $\mathcal{A}_i^\phi$ is unnecessary (in this case $\mathbf{A}_i=\mathcal{A}_i^\phi$). However, PSD  $\mathbf{A}_i$ is rare. For instance, in the dataset AIDS, the number of PSD $\mathbf{A}_i$ is **ZERO**.
>
> **To Q3:**
>
> We used the sklearn function: sklearn.utils.extmath.randomized$\_$svd(M, n$\_$components=d, random$\_$state=0). This can be found in the code of the supplementary material. We added these details to the revised paper.
>
> **To Q4:**
>
> We assume $n_1=n_2=n$ so as to simplify the notations in many formulas or results such as (8), (12), (16), Theorem 2.3, and Theorem 2.4. In the definition of MMFD, i.e. Definition 2.1, we can replace $n$ with $n_1$ and $n_2$ without influencing the meaning of the definition, since it is based on the mean vectors.
> Therefore, we claimed "without loss of generality". We will use $n_1$ and $n_2$ throughout the paper if you think it is necessary.
>
> Regarding the MMFD between two complete graphs of different sizes, we provide the following theoretical proof.
>
> A complete graph is a graph in which each vertex is connected to every other vertex. Therefore, for two complete graphs, their self-looped adjacency matrices are $\mathbf{A} _i=[1] _{n _i\times n _i}$, $i=1,2$.
> $\mathbf{A} _i$ are PSD and rank-1. We have $\boldsymbol{\Phi} _i=s\cdot[1] _{1\times n _i}$, where $s$ is $-1$ or $1$. That means $\boldsymbol{\mu} _i=-1$ or $+1$. The rotation matrix $\mathbf{R} _{12}$ is now just a scalar equal to $-1$ or $+1$. Therefore, $\min _{\mathbf{R} _{12}\in\mathcal{R}} \Vert \boldsymbol{\mu} _1-\mathbf{R} _{12}\boldsymbol{\mu} _2 \Vert\equiv 0$, for any $(n _1,n _2)$.
>
> We added the above proof to the revised paper.
>
> **To Q5:**
>
> Sorry for the confusion. The superscript 0.5 is the square root. We visualize the square root of the distance to make the visual difference clearer. As you know, in many studies, people use a log scale to show the results. Here, we use a square-root scale.
>
> **Thank you again for reviewing our paper. We are looking forward to your feedback.**

---

> > ### Comment · Reviewer_8oX9 · 2025-04-05
> >
> > I responded the authors before my rebuttal acknowledgement, but it seems not visible to the authors. I restate it here. Sorry for that.
> >
> > Thank the authors' response. Many concerns have been addressed except the major one.
> >
> > The non-uniqueness of $\Phi_i$ is algebraically true, but it doesn't mean that any factorization of $\mathcal{A}_i$ makes sense. It seems more reasonable to find the most suitable factorization for each $\mathcal{A}_i$ and measure the distance based on these specific factorizations. The minimum mean factor distance actually implies a "rotational equivalence" in the representations. Even though to measure the difference of two sets of representations (I don't use the word "distance" here because distance means "minimum", but I think we just need to measure the difference of two graphs), we still have several options. Minimum distance is one of them, but it is hard to say which one is a natural principle. Moreover, at the very least, the final format of MMFD only relates to the sum of entries of $\mathcal{A}_i$, which seems quite weird. Imagine that $\mathcal{A}_i$ is simply an adjacency matrix that is symmetric. Does it mean that the MMFD of two graphs that have equal size and the same number of edges is 0? Sometimes, excessive simplification is not necessarily a good thing, since it is possible to lose useful information.

---

> > > ### Author Response · Authors · 2025-04-05
> > >
> > > Dear Reviewer 8oX9,
> > >
> > > Thank you so much for your response.
> > >
> > > * Regarding the options of distance measure, indeed, it is hard to say which one is a natural principle, but a minimum distance is meaningful, and we are willing to rephrase the presentation. As you can see, the proposed MMFD makes sense theoretically and performs in all experiments.
> > >
> > > * The final format of MMFD is not related to the sum of entries of the original adjacency matrix $\mathbf{A}$. Instead, it is related to $\mathcal{A}^\phi$, which is $\mathbf{PSD}$ and different from $\mathbf{A}$. PSD $\mathbf{A}$ is rare; for instance, in the four datasets of Table 3 (8,712 graphs), the total number of PSD $\mathbf{A}$ is only 14.
> > >
> > > * We convert the graph comparison problem into a discrete distribution comparison problem, where each sample $\mathbf{z}$ corresponds to a node in the graph. Specifically,  we construct the PSD matrix $\mathcal{A}^\phi$ so as to obtain the meaningful feature representation $\phi(\mathbf{z})$, and then compare the means of $\phi(\mathbf{z})$ between two graphs, leading to a distance measure between two distributions.
> > >
> > > * MMFD is theoretically founded, although the final format is quite simple. Regarding your question about two graphs with an equal number of nodes and the same size, if the original self-looped adjacency matrices are both PSD, yes, the distance is zero. An example is that MMFD between two **complete graphs** is always zero. However, for graphs with non-PSD adjacency matrices, which is usually the case, the distance is often not zero.
> > >    * For instance, in Table 2 of our paper,  $G_5$, $G_6$, and $G_7$ have the **same number of nodes and the same number of edges**, but their MMFDs are all **not zero**. This sufficiently demonstrated the ability of MMFD to distinguish highly similar graphs.
> > >    * It should be emphasized again that the usefulness of MMFD, including the initial definition and the final format, is based on PSD $\mathcal{A}_i^\phi$. By the way, a symmetric matrix is not necessarily a PSD matrix.
> > >
> > > * Besides the basic MMFD, we have an extension MFD introduced in Section 2.8, which is more complex and is able to utilize more information. As shown in Table 3 of our paper, MFD is more effective than MMFD.
> > >
> > > We are very grateful for your thoughtful discussion. It appears that your concern is not directly related to the effectiveness, theoretical soundness, and completeness of our work, whereas the other three reviewers do not have concerns regarding these aspects. Please do not hesitate to let us know if your concern remains.
> > >
> > > Sincerely,
> > >
> > > Authors
> > >
> > > April 05
> > >
> > > \-----------------------------
> > >
> > > \-----------------------------
> > >
> > > \-----------------------------
> > >
> > > Hi Reviewer 8oX9,
> > >
> > > Did our previous explanation resolve your concern?
> > > * Although the computation of MMFD is very simple, it is theoretically founded.
> > > * MMFD outperformed many complicated methods such as WL kernel, graph edit distance, gromov wasserstein distance, and GNNs in the experiments.
> > > * In addition to MMFD, we have provided a more complex extension called MFD, which outperformed MMFD in the experiments.
> > >
> > > Sincerely,
> > >
> > > Authors

---

### Decision · Program_Chairs · 2025-05-01

**Decision:**

Accept (poster)

**Comment:**

This paper presents efficient approaches for evaluating graph similarities using the minimum mean factor distance. As all reviewers have provided positive evaluations, I recommend acceptance of the paper.